# Bridging Gene Expression and Text: LLMs Can Complement Single-Cell Foundation Models

## Abstract

Single-cell foundation models such as scGPT represent a significant advancement in single-cell omics, with an ability to achieve state-of-the-art performance on various downstream biological tasks. However, these models are inherently limited in that a vast amount of information in biology exists as text, which they are unable to leverage. There have therefore been several recent works that propose the use of LLMs as an alternative to single-cell foundation models, achieving competitive results. However, there is little understanding of what factors drive this performance, along with a strong focus on using LLMs as an alternative, rather than complementary approach to single-cell foundation models. In this study, we therefore investigate what biological insights contribute toward the performance of LLMs when applied to single-cell data, and how these models can complement single-cell foundation models to improve upon their performance. We first conduct a series of interpretability and ablation tests which show that LLMs leverage marker gene knowledge and simple gene expression patterns, contributing to their competitive performance. We then introduce scMPT, a proof-of-concept model which combines single-cell representations from LLMs and single-cell foundation models, demonstrating synergies between these representations through stronger, more consistent performance across datasets and tasks. We also experiment with alternate fusion methods, which highlight the potential of combining specialized reasoning models with scGPT to improve performance. This study ultimately showcases the potential for LLMs to complement single-cell foundation models and drive improvements in single-cell analysis.

## 1 Introduction

Single-cell foundation models, such as scGPT (Cui et al., 2024), have seen a surge in recent interest due to their ability to be adapted to achieve state-of-the-art performance on a variety of biological tasks. However, these models have inherent limitations. A vast amount of knowledge in the field of biology is represented as text, but these single-cell foundation models are trained only on gene expression data, and have no way to use this information. There has therefore been interest in applying large language models (LLMs) to single-cell transcriptomics, as many of these models have a large amount of pretrained knowledge which encompasses this knowledge of biology. LLMs could potentially leverage this knowledge to drive improvements in important tasks in single-cell analysis. Either as an alternative approach, circumventing the need to curate massive amounts of data to train new single-cell foundation models, or as a complementary approach, merging the knowledge and capabilities of LLMs and single-cell foundation models to improve performance over either.

A popular approach for enabling LLMs to work with single-cell data is converting this data to simple text sequences (i.e. cell sentences), which are encoded to generate representations that intend to encapsulate the knowledge obtained during training, or parametric knowledge, of these models. This method has yielded promising results that are competitive with dedicated foundation models on certain single-cell analysis tasks such as cell type classification (Chen & Zou, 2023) (Choi et al., 2024) (Levine et al., 2023) (Fang et al., 2024) (Rizvi et al., 2025). However, what knowledge is being captured in these representations, as well as how to leverage potential synergies between them and representations generated by single-cell foundation models remains largely unexplored. These questions are necessary to understand how LLMs can meaningfully improve single-cell analysis and address the shortcomings of single-cell foundation models.

In this work, we use interpretability methods along with an ablation study to elucidate what knowledge of biology and other factors LLMs leverage when applied to single-cell analysis. We then explore how these models can be used to complement single-cell foundation models, improving performance and demonstrating synergies between representations. Our key contributions are:

1. We find that LLMs capture biological insight, and specifically knowledge of marker genes, as well as simple but effective gene expression patterns when generating representations of single-cell data. Our results further suggest this is facilitated through their leveraging of parametric knowledge of biology, even in a setting where the input representations used are simple cell sentences, and the LLMs used are off-the-shelf text encoders.

2. We find that fusion with LLMs can improve upon the performance of single-cell foundation models, indicating that cell representations derived from text and single-cell data are complementary. We introduce scMPT, which leverages synergies between representations generated by scGPT and an Ember-V1 text encoder, enabling better overall performance for cell-type classification and disease phenotype prediction. Furthermore, our experiments with alternate fusion methods suggest that reasoning capabilities can enable LLMs to more effectively complement single-cell foundation models, highlighting a promising direction for future work.

## 2 RELATED WORK

### 2.1 LARGE LANGUAGE MODELS

LLMs have demonstrated strong performance and versatility across a variety of domains and tasks. They have been shown to perform well on classification, question-answering, fact retrieval, and more, even without fine-tuning (Gallegos et al., 2024). These models, typically based on the Transformer architecture (Vaswani, 2017), consist of millions or even billions of trainable parameters, and are pre-trained with vast amounts of language data which often encompasses many domains, facilitating this versatility (Zhang et al., 2024a).

LLMs are often used to generate text in an autoregressive fashion, or to generate representations of text in the form of embeddings that can be used for a variety of downstream tasks. However, models used for each of these tasks are generally quite different, with LLMs designed for text generation typically employing a different type of architecture than text embedding models (Zhang et al., 2024a). We will therefore study each type of model separately.

### 2.2 SINGLE-CELL FOUNDATION MODELS

Inspired by the success of LLMs, single-cell foundation models such as scGPT have been developed that display broad capabilities across many biological tasks such as cell type annotation and multi-batch integration (Chen & Zou, 2023) (Cui et al., 2024). scGPT is, like most LLMs, based on the transformer architecture. Key to its development was curating and pre-training on a massive amount of single-cell data, specifically from over 33 million cells from CELLxGENE (Cui et al., 2024) (Megill et al., 2021).

### 2.3 APPLYING LARGE LANGUAGE MODELS TO SINGLE-CELL ANALYSIS

To enable large language models to work with single-cell data, existing studies generally represent this data as text. Perhaps the most common representation used, which we will focus on in this study, is the "cell sentence"; a textual sequence which lists gene names in descending order of expression level for a given cell. For example, *"A cell with genes ranked by expression: RAB3B MT-CO1 CHN1 HNRNPA1P40 SYT1....."*. The Cell2Sentence paper demonstrated that conversion to this representation incurs minimal information loss. This was accomplished through training a linear model to accurately predict gene expression from gene rank, and motivates our focus on this method (Levine et al., 2023). Studies that use LLMs with cell sentences for single-cell analysis broadly fit into two categories; those that use generative models, and those that use embedding models.

Studies that explore the use of generative models include Cell2Sentence (Levine et al., 2023) and its extension "Scaling Large Language Models For Next-Generation Single-Cell Analysis" which presents Cell2Sentence-Scale (Rizvi et al., 2025), as well as "How do Large Language Models

understand Genes and Cells" (Fang et al., 2024). The results presented in this last study fell short of scGPT on tasks such as cell type annotation, despite using cell sentences to fine tune large LLMs with up to 13 billion parameters. While the performance of Cell2Sentence and Cell2Sentence-Scale is much more competitive with single-cell foundation models such as scGPT, they require curating large single-cell datasets for a specialized training process which involves multiple stages and is computationally expensive. However, ideally this would not be necessary when working with LLMs due to the knowledge of biology they obtain during pre-training, which can potentially enable "off-the-shelf" usage, or a merging of this knowledge with single-cell foundation models that have already undergone specialized training with single-cell data.

Studies that use embedding models include GenePT (specifically, the GenePT-s approach) (Chen & Zou, 2023), and CELLama (Choi et al., 2024). Both papers reported performance that was competitive with scGPT on tasks such as cell type classification in a zero-shot setting. However, a limitation of these works is that they provide little justification for their selection of embedding models, despite these models often varying widely in performance on different tasks (Muennighoff et al., 2022).

Another limitation of these studies on applying LLMs to single-cell analysis is that they primarily focus on how LLMs can be used as an alternative to single-cell foundation models, rather than in a complementary fashion. The GenePT paper does notably experiment with an ensemble approach that aggregates the nearest neighbours from GenePT-s, scGPT, as well as their GenePT-w method to make a final cell type prediction (Chen & Zou, 2023). However, fusion at such a late stage ignores possible synergies between different modalities (Steyaert et al., 2023).

## 3 METHODS

### 3.1 DATA COLLECTION AND TRANSFORMATION

We focus our experiments on the datasets that were used to evaluate the cell type classification and clustering performance of GenePT (Chen & Zou, 2023), as well as the subsample of the Tabula Sapiens dataset (Consortium* et al., 2022) used to evaluate CELLama (Choi et al., 2024). For each dataset, we use the same train/test split as each of these respective works to evaluate performance on downstream tasks. Cells were represented using cell sentences following the approach of GenePT-s (Chen & Zou, 2023), where gene names are listed in descending order of expression level, omitting genes with zero counts. These cell sentences are then passed to text encoders to generate cell embeddings, or to generative LLMs for cell type classification through autoregressive generation.

### 3.2 CELL EMBEDDING APPROACHES

In general, text encoders can vary greatly in performance on tasks such as classification and clustering. To determine what LLM of this type to use for our experiments, we therefore test a variety of pre-trained models, selected based on Massive Text Embedding Benchmark (MTEB) performance (Muennighoff et al., 2022). We use the same experimental design and metrics as the GenePT paper to evaluate classification and clustering performance (Chen & Zou, 2023). For example, using a 10-nearest neighbours method for zero-shot cell type classification. Details can be found in Appendix A.1. Overall, we find that Ember-V1 outperforms text embedding models used in previous work for generating cell embeddings from cell sentences, motivating us to select this encoder for our experiments.

We then experiment with training a small multilayer perceptron (MLP) on top of Ember-V1. This setup has the potential to improve cell type classification performance over k-nearest neighbours with minimal training, but more importantly, the differentiability of the MLP facilitates a wider range of interpretability methods. We leave the text encoder frozen during training to reduce computational cost. Building on this idea of training an MLP on top of Ember-V1, we then train a multimodal network for cell type classification on top of Ember-V1 and scGPT, which we coin scMPT. scMPT combines extracted features from each encoder to leverage potential representational synergies, evaluated through performance changes relative to the individual encoders. Both encoders are left frozen during training to reduce computational cost, and maintain the domain specific knowledge encoded in each model. We report accuracy along with macro-weighted precision, recall, and F1 score for all datasets, comparing to each of scMPT's component encoders with MLP classification heads as a baseline. The architecture of scMPT takes inspiration from previous studies

on multimodal deep learning (Kwak et al., 2023) (Miller et al., 2020), and is presented in Figure 1 below. The simplicity of this architecture and use of representation-level fusion help ensure that any performance improvements are primarily driven by synergies between representations, rather than scMPT having a greater capacity than the classification heads used for the component encoders.

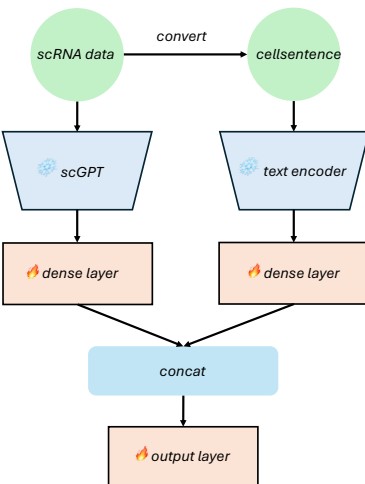

Figure 1: The scMPT model architecture. Cell embeddings from scGPT and a text encoder are fed into dense layers. The output of these dense layers is then concatenated, and fed into an output layer which predicts final cell type. Encoders are left frozen, while dense and output layers are trainable.

### 3.3 GENERATIVE APPROACHES

As an alternative multimodal approach to scMPT, we then investigate whether generative LLMs can be used to complement scGPT, leveraging the domain specific knowledge encoded in each model to improve classification performance. Our pipeline for all generative LLMs tested is as follows: For each cell we want to classify, we first determine the three most likely cell types using scGPT and the 10-nearest neighbour method of cell type classification previously described. We then prompt the generative LLM to determine which of these cell types is most likely correct given the cell sentence for the cell of interest. We prompt the model to select between the top three cell types based on scGPT's strong top-3 accuracy shown in Appendix A.3 Table 24. We also provide instructions in the prompt to pick the first cell in the list, which is the most likely class according to scGPT, if the LLM is uncertain. Additional details can be found in Appendix A.3.2, with the prompt in Figure 5.

For this experiment, we test both standard LLMs, and reasoning models, which are specifically trained for improved reasoning using strategies such as reinforcement learning (Guo et al., 2025; OpenAI, b). The standard LLMs evaluated include GPT-4o, and DeepSeek-V3. The reasoning models evaluated, o3-mini and DeepSeek-R1, are taken from the same model families as the standard LLMs, enabling us to evaluate the impact of reasoning capabilities.

### 3.4 INVESTIGATING WHAT FACTORS CONTRIBUTE TOWARD LLMS' PERFORMANCE

To investigate the factors contributing to the competitive performance of LLMs, specifically focusing on text encoders, we first investigate what features from cell sentences are focused on by Ember-V1 when predicting cell type. We focus on text encoders as this form of LLM has proven to be competitive with scGPT on certain single-cell analysis tasks (ie. as shown in Appendix A.1), and models in this class tend to be small enough that a wider range of interpretability methods are computationally feasible. We employ two interpretability techniques. The first, integrated gradients, is a gradient based method which has seen significant recent adoption in the biomedical domain, including for interpreting language models (Sundararajan et al., 2017) (Talebi et al., 2024). The second, Local Interpretable Model Agnostic Explanations (LIME) is a model agnostic interpretability method that has also seen significant usage in this domain (Ribeiro et al., 2016) (Wu et al., 2023) (Laatifi et al., 2023). These methods notably provide more insightful explanations than examining

attention weights (Lopardo et al., 2024). We apply both interpretability techniques on the model that consists of an MLP trained on top of Ember-V1. We focus on this model rather than the setup that uses k-nearest neighbours since the MLP is differentiable, facilitating the usage of integrated gradients. We limit our analysis to cell types that are specific and have clearly defined names. For each dataset and each interpretability method, we calculate feature attributions for ten cells of each type (or all cells if there are less than ten in the test set). We then sum attributions across the ten cells of each type, and examine the top ten genes with the highest positive attribution scores to determine what genes influence the model to predict a cell is of a given type.

We also conduct a series of ablation tests to better understand what factors contribute to the performance of text encoders. We ablate each major element of biological information in the cell sentences that was derived from the raw single-cell data: the gene names, and the ordering of gene names which was based on expression level. We then observe how this impacts clustering and classification performance. Specifically, for our first ablation test, we replace gene names in cell sentences with unique hashes generated using the SHA-256 algorithm, truncated to ten characters. For our next ablation, we investigate the effect of randomly shuffling the order of gene names within cell sentences. We begin by shuffling all gene names to fully ablate the step of ordering these genes by expression level. To better understand the impact of order, we then experiment with shuffling only gene names that are in the input context length of the text encoder, then only the top 10% of in-context genes. Finally, we investigate the effect of combining these gene name and order ablations.

## 4 RESULTS

### 4.1 LANGUAGE MODELS EFFECTIVELY LEVERAGE MARKER GENE KNOWLEDGE AND SIMPLE GENE EXPRESSION PATTERNS

To determine what factors contribute toward the strong performance of language models, and in particular text encoders, we conduct a series of interpretability and ablation tests.

The interpretability methods we use are integrated gradients, and LIME. For this analysis we focus on the Pancreas and Aorta datasets, given the strong performance of Ember-V1 on them. We apply both interpretability methods to a model consisting of an MLP trained on top of a frozen Ember-V1 encoder to determine what parts of cell sentences contribute toward the prediction of certain cell types. Specifically, attribution scores were calculated for different gene names, which show how much those gene names contributed toward predicting that a cell was of a given type. Early in this analysis, we noticed that several of the genes corresponding to top attribution scores were marker genes. We therefore used the PanglaoDB marker database (Franzén et al., 2019) to investigate what marker genes were represented in the lists of top attribution scores for each cell type, calculated through aggregating results from multiple cells of a given type for a more global level of insight. As shown in Table 1, for each cell type in the Pancreas dataset, both methods had multiple markers within the top ten genes with the highest attribution scores. There was also at least one and often multiple markers highlighted by both interpretability methods for each cell type, with this agreement more strongly indicating these marker genes were focused on by the model. For the Aorta dataset (see Table 32 in Appendix A.5), LIME once again had multiple markers within the top ten most important genes for each cell type. While integrated gradients had less, it is notable that the markers highlighted by integrated gradients were almost an exact subset of the markers highlighted by LIME, suggesting these markers were focused on by the model. Ultimately, the representation of marker genes among the genes with the highest attribution scores for each cell type, along with the level of agreement between interpretability methods indicates that the model focuses on marker genes when predicting cell type, contributing to its performance being competitive with scGPT.

For our ablation tests, we experiment with ablating the major elements of biological information in cell sentences; the gene names, which are replaced with truncated SHA-256 hashes, and their order, which is shuffled for different proportions of the cell sentence. We report how each of these ablations, along with their combination, affect performance on the Aorta dataset for Ember-V1 in Table 2, and ada-002 in Appendix A.4 Table 27, using the k-nearest neighbours setup for cell type classification. Note that to be able to compare the effect of ablations on each encoder's performance, we truncate the cell sentences encoded by ada-002 to match Ember-V1's input context length.

We find that both encoders, and especially ada-002, had only a moderate drop in performance when ablating gene names, despite the truncated SHA-256 hashes they are replaced with having no biolog-

Table 1: Marker genes in top 10 gene attributions for Ember-V1 encoder + MLP with different interpretability methods - Pancreas dataset.
*(Markers that were highlighted by both interpretability methods in bold)*

| Cell Type | LIME | Integrated Gradients |
|---|---|---|
| **Alpha** | **GCG**, **TTR**, CRYBA2, TM4SF4, LOXL4 | **GCG**, **TTR**, NEUROD1, GC, ALDH1A1, SLC30A8 |
| **Beta** | INS, **IAPP**, **ADCYAP1** | **ADCYAP1**, **IAPP** |
| **Gamma (PP)** | **PPY**, ISL1 | **PPY**, NEUROD1 |
| **PSC** | **COL6A3**, FN1, TIMP1 | COL1A1, COL1A2, **COL6A3**, COL3A1 |
| **Ductal** | SERPING1, **KRT19**, MUC1 | **KRT19**, MMP7, SERPINA3 |
| **Endothelial** | **PLVAP**, **RGCC**, PODXL | **PLVAP**, **RGCC**, IGFBP7 |
| **Epsilon** | **GHRL**, **S100A6**, **SPINK1**, ACSL1 | **GHRL**, **SPINK1**, **S100A6**, HMGCS2 |
| **Mast** | **TPSAB1**, CPA3 | **TPSAB1**, ICT4S |
| **Acinar** | **PRSS1**, **CXCL17**, **PRSS3**, **REG1A** | **PRSS1**, **PRSS3**, **REG1A**, CELA3A, **CXCL17**, CELA2A |
| **Delta** | **SST**, **RBP4**, LEPR, PCSK1 | **SST**, **RBP4** |

ical meaning. To confirm this is not because of unique characteristics of the Aorta dataset, we also see how this ablation affects the cell type classification performance of Ember-V1 on other datasets, finding a consistently moderate drop as seen in Appendix A.4 Tables 30 and 31. A potential explanation for this is that cells of the same type will generally have more similar cell sentences in terms of gene name order, and since gene names will always be mapped to the same hash, they will also have more similar ablated cell sentences. We find that a per-instance gene name ablation, where each instance of each gene name is replaced with a random unique hexadecimal string resembling a SHA-256 hash, leads to a near total loss of predictive performance, supporting this explanation. This is seen in Table 2 as well as Appendix A.4 Tables 28 and 29. The results of these ablations suggest that although gene names and their corresponding biological signal have a moderate contribution to encoder performance, the simple similarity between word order and presence in cell sentences (or sequential and lexical similarity), even without considering semantics, also has importance.

For our order ablations, we first find that shuffling all gene names for each cell results in a near total loss of predictive performance for both encoders since in this scenario, the cell sentence for each cell becomes a completely random list of gene names. In contrast, we find that only shuffling gene names within the context length of Ember-V1 leads to a moderate drop in performance. Note that the context length of Ember-V1 can only contain a small subset of the top expressed genes for a given cell, such as approximately 152 genes, on average, for the Aorta dataset. In this ablation, the cell sentences for each cell will continue to contain these top expressed genes, just in a different order, potentially explaining the smaller drop in performance when comparing to the result of shuffling all gene names.

Shuffling only the top 10% of in context gene names leads to a disproportionately large drop in performance when compared to the drop in performance when shuffling all in context gene names. This is shown in Table 2, where it accounts for more than 50% of the drop in accuracy and F1 score for Ember-V1 on the Aorta dataset relative to shuffling all in context gene names. We observe a similar trend for other datasets, as can be seen in Appendix A.4 Table 28 and 29, as well as for ada-002, as can be seen in Table 27. This suggests that the model's performance is heavily reliant on this small subset of the cell sentence. To further investigate this, we experiment with using shortened cell sentences which include only the top 10%, 20% and 50% of in context genes by expression level, and seeing how this impacts performance for Ember-V1. As shown in Figure 6 in Appendix A.4, we find that performance even when using only the top 10% of genes is relatively strong, and competitive with ada-002 performance when using all genes. On the datasets where Ember-V1 originally outperformed scGPT substantially (the Pancreas and Bones datasets), we observe that it still outperforms scGPT substantially even when using only the top 10% of in context genes, as

shown in Figure 7 and 8 of Appendix A.4. Furthermore, we note diminishing returns as more of the cell sentence is used, despite the relatively short context length of Ember-V1.

Finally, we find that combining ablations by ablating and then shuffling all gene names within the context length of Ember-V1 leads to a severe drop in performance. However, some performance is still maintained, with Ember-V1 still achieving an F1 score of over 0.2 on two of the three datasets where this ablation was tested, as seen in Table 2, and Table 28 in Appendix A.4. Note that cells of the same type will generally have higher overlap between their top expressed genes, and by extension, higher overlap between the shuffled random hashes present in their cell sentences in this scenario. The remaining performance for this ablation suggests that encoders leverage this lexical similarity between ablated cell sentences, generating cell embeddings that are generally at least slightly more similar for cells of the same type.

Overall, our ablations suggest that Ember-V1 leverages simple gene expression patterns, contributing to its performance. This is demonstrated by the importance the model places on the top 10% of in context genes and their order, and the diminishing returns as more of the cell sentence is used. It is further supported by the only moderate drop in performance when ablating gene names, and the retaining of some performance even when combining gene name and in context order ablations, which suggest that along with biological meaning, the model leverages sequential and lexical similarity between cell sentences.

Table 2: Ablation test results – Aorta Data (Ember-V1 – zero-shot / KNN).

| Metric | No Ablations (Baseline) | Gene Name Ablation | Order Ablation (All Genes) | Order Ablation (In Context) | Order Ablation (Top 10% In Context) | Gene Name + Order Ablation (In Context) | Gene Name Per-Instance Ablation |
|---|---|---|---|---|---|---|---|
| **Accuracy** | **0.906** | 0.830 | 0.334 | 0.856 | 0.877 | 0.615 | 0.311 |
| **Precision** | **0.910** | 0.816 | 0.092 | 0.902 | 0.867 | 0.286 | 0.085 |
| **Recall** | **0.800** | 0.563 | 0.093 | 0.623 | 0.706 | 0.221 | 0.087 |
| **F1** | **0.841** | 0.605 | 0.079 | 0.681 | 0.754 | 0.216 | 0.077 |
| **k-means ARI** | **0.535** | 0.360 | 0.000 | 0.506 | 0.372 | 0.092 | -0.002 |
| **k-means AMI** | **0.597** | 0.390 | 0.000 | 0.529 | 0.515 | 0.081 | 0.000 |

Ultimately, our interpretability and ablation tests indicate that text embedding models effectively leverage both marker gene knowledge and simple gene expression patterns. These factors contribute to the models' competitive performance with scGPT for cell type classification and clustering.

### 4.2 SCMPT AND GENERATIVE LLM PIPELINES ENABLE LLMS TO COMPLEMENT SCGPT, IMPROVING PERFORMANCE AND DEMONSTRATING REPRESENTATIONAL SYNERGIES

Finally, we investigate how LLMs can be leveraged to complement single-cell foundation models, and whether single-cell representations from LLMs and single-cell foundation models exhibit synergy.

We first investigate representation-level fusion with Ember-V1 and single-cell foundation models to leverage potential synergies. For each dataset, we use the train split to train a simple multimodal neural network on top of Ember-V1 and scGPT, which are left frozen. We then evaluate cell type classification performance on the test split. To provide a baseline, we also evaluate the performance of training an MLP on top of each individual encoder. We report results for the Pancreas dataset in Table 3 and results for other datasets in Appendix A.2. We observe that the fusion model, which we coin scMPT, performs competitively with, and often better than the best of the two encoders on each dataset. The performance of scMPT is notably strong enough that it is even competitive with full fine tunes of scGPT, based on results reported in the original scGPT paper (Cui et al., 2024). We additionally find that scMPT performs better than scGPT and Ember-V1 when trained and tested on disease phenotype prediction, as shown in Table 19 and 20 of Appendix A.2, indicating that its effectiveness generalizes beyond cell type classification. We further observe the generalizability of

this framework through using the Geneformer (Theodoris et al., 2023) single-cell foundation model in place of scGPT within scMPT, and finding scMPT once again yields performance gains over its component models on both tasks, as seen in Table 21 and 22 of Appendix A.2. We also notably observe that simply training an MLP on top of a frozen encoder can improve cell type classification performance over k-nearest neighbours. This is shown in Figure 2 and Figure 3 (see Appendix A.2).

Table 3: Cell type classification performance of scMPT vs. unimodal models on the Pancreas dataset. scMPT results reported as Mean (Standard Deviation) across 5 runs.

| Model | Accuracy | Precision | Recall | F1 |
|---|---|---|---|---|
| **scMPT** | 0.962 (0.0019) | **0.764 (0.0017)** | **0.752 (0.0037)** | **0.745 (0.0039)** |
| **scGPT (full fine-tune - reported)** | 0.968 | 0.735 | 0.725 | 0.718 |
| **Ember-V1 + MLP** | **0.974** | 0.6815 | 0.694 | 0.684 |
| **scGPT (from-scratch - reported)** | 0.936 | 0.665 | 0.668 | 0.622 |
| **scGPT + MLP** | 0.865 | 0.625 | 0.614 | 0.592 |

As an alternative to scMPT, we next investigate how generative LLMs can be used to complement scGPT. Specifically, through using these LLMs to guide the predictions of scGPT, narrowing down the top three cell types predicted as most likely for a cell of interest to a final prediction based on the cell's cell sentence. We report results on subsets of 100 cells (to limit costs) from each of the Pancreas, Myeloid, and MS test sets, and include the accuracy of scGPT and o3-mini as a baseline, in Table 4 below. Our method that uses o3-mini to complement scGPT performs particularly well, outperforming scGPT across all three datasets, and greatly outperforming o3-mini alone on the Myeloid and MS datasets while still performing competitively on the Pancreas dataset. In general, reasoning models combined with scGPT outperformed standard LLMs combined with scGPT. We also find that reasoning models outperform standard LLMs when these models are used for cell type classification without scGPT, as shown in Appendix A.3.1 Table 23, further demonstrating the benefit of reasoning models and reasoning capabilities in this setting.

Table 4: Accuracy of scGPT, o3-mini, and fusion method with different generative LLMs reported as Mean (Standard Deviation) across 3 runs. Reasoning models underlined.

| Model | Pancreas Data | Myeloid Data | MS Data |
|---|---|---|---|
| scGPT | 0.78 | 0.51 | 0.75 |
| o3-mini | **0.980 (0.010)** | 0.393 (0.015) | 0.473 (0.012) |
| DeepSeek-V3 + scGPT | 0.920 (0.000) | 0.503 (0.0058) | 0.687 (0.0058) |
| DeepSeek-R1 + scGPT | 0.930 (0.000) | 0.520 (0.0265) | 0.733 (0.0058) |
| gpt-4o + scGPT | 0.920 (0.000) | 0.480 (0.0265) | 0.690 (0.0265) |
| o3-mini + scGPT | 0.930 (0.000) | **0.530 (0.0100)** | **0.763 (0.0058)** |

Overall, we find methods that use LLMs to complement scGPT can yield cell type classification performance competitive with, and often better than the more performant of the two component models. scMPT, which uses Ember-V1 to complement scGPT through representation-level fusion, performs particularly well, highlighting representational synergies. Using reasoning models to complement scGPT also performs well, outperforming the use of standard generative LLMs in this setting.

## 5 DISCUSSION

LLMs have shown great potential in single-cell analysis, often performing competitively with dedicated state of the art foundation models. In this study, we obtain an understanding of the biological insight and other factors contributing to this performance, and explore whether this leads to single-cell representations that are synergistic with single-cell foundation models.

Through interpretability methods, we find that Ember-V1 focuses on marker genes when predicting cell type. This is intriguing, as focusing on markers is a common approach for both automatic

and manual cell type annotation (Clarke et al., 2021). One important limitation of our analysis is that integrated gradients and LIME are local interpretability methods. While our strategy of aggregating attribution scores across many examples can help get a more global understanding of the model's behaviour, and has been used in previous work (Talebi et al., 2024), this method is still limited. We therefore take inspiration from the analysis of Liétard et al. (2021) to get a more global understanding of Ember-V1 and its knowledge of marker genes. For each dataset used with the other interpretability methods, we compute the average cosine similarity between Ember-V1 embeddings of the top marker gene name for each cell type from PanglaoDB (Franzén et al., 2019) and other marker genes from the same cell type, and compare this to the average cosine similarity between embeddings of these top markers and marker genes of different cell types. We then average results over all cell types. As shown in Appendix A.5 Table 33, we find that the cosine similarities to marker genes of the same cell type (Intra-Similarity) are higher, providing further evidence that the model is able to leverage knowledge of marker genes. In Appendix A.3.3 Table 25 and Table 26, we conduct a similar test for generative models, which indicates that they are also able to leverage knowledge of marker genes. Ultimately, these results, in tandem with the findings of our interpretability tests, and the moderate drop in performance observed when ablating gene names, strongly suggest that LLMs, and in particular Ember-V1, leverage knowledge of biology, and specifically knowledge of marker genes obtained during training when applied to single-cell data.

Ember-V1's ability to leverage simple gene expression patterns, based on our ablation studies, is another intriguing finding. As with marker genes, the model may have been exposed to these patterns during training. With that said, the model's leveraging of lexical and semantic similarity between cell sentences also contributes toward this ability. Being able to exploit these simple patterns is potentially advantageous, since, for example, the highest expressed genes and their rank for different cells of the same type may be quite similar. However, the model may not be effective in leveraging more complex gene expression patterns, as suggested by the diminishing returns observed when more than 10% of in-context genes are used. In contrast, single-cell foundation models are able to leverage relatively complex patterns, such as a general understanding of even long-range gene-gene interactions Cui et al. (2024) Yang et al. (2022), suggesting these models may be more robust in settings where cell identity is primarily defined by these more complex relationships. This discrepancy between the strengths of the two different types of model potentially contributes to the large performance gaps between them across datasets. For instance, Ember-V1 outperformed scGPT on the Pancreas and Bones datasets, whereas scGPT outperformed Ember-V1 on the Multiple Sclerosis dataset. Based on these strengths, it is difficult to know in advance which model would be better suited for aiding in the analysis of a given single-cell dataset. This motivates the development and use of fusion methods that can effectively leverage the strengths of both types of models.

The fusion methods evaluated in this study performed well, especially scMPT, highlighting synergies between representations. On datasets where both modalities yielded similar performance individually, the performance of scMPT was generally modestly stronger than either individual modality. Furthermore, on datasets where one modality yielded poor performance, scMPT still performed well. scMPT's strong performance generalized across both tasks evaluated, as well as when Geneformer was used in place of scGPT. The method that combined reasoning models with scGPT also performed well, with the improvement in performance over standard generative LLMs in this setting suggesting that reasoning capabilities can enable LLMs to more effectively complement single-cell foundation models. Although this setup did not achieve the same level of accuracy as scMPT, it should be noted that scMPT uses a deeper level of fusion, which is specifically at the representation level, and includes a component that is fine-tuned.

One limitation of this work is the scope of evaluation for fusion methods. An important direction for future research would therefore be to test the performance of LLMs, scGPT, and fusion methods on more datasets and tasks to gain a more comprehensive understanding of when each type of model may perform better. Another direction for future research is the development of more advanced fusion methods. For example, large multimodal models are a promising direction due to their strong multimodal classification performance (Alayrac et al., 2022). It would be particularly interesting to explore the development of large multimodal models with reasoning capabilities, as our results highlight the potential value of reasoning in this setting. Finally, another limitation of this study is that only cell sentences were focused on as textual representations of single-cell data. In future work, it will be valuable to investigate whether there are other textual representations of single-cell data that can drive stronger performance, for instance, taking inspiration from emerging work in the

single-cell foundation model space which explores moving beyond representing cells as sentences in order to capture cell-cell relationships Wen et al. (2023).

## 6 REPRODUCIBILITY STATEMENT

References to all datasets and pre-trained models used are provided in this work. The architecture of any models trained is specified, with additional details for scMPT provided in Appendix A.2. Generative LLM prompts and exact model versions used can be found in Appendix A.3.2. Dataset splits are discussed in Section 3.1.

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

# A    APPENDIX

**Contents of Appendix**

## A.1 ZERO-SHOT PERFORMANCE OF VARIOUS ENCODERS - DETAILS AND RESULTS

To select an encoder for our main experiments, we test a variety of pre-trained LLMs for generating cell embeddings from cell sentences. Encoders were specifically selected through looking at classification performance on the Massive Text Embedding Benchmark (MTEB) (Muennighoff et al., 2022) (https://huggingface.co/spaces/mteb/leaderboard), filtering out encoders that are proprietary, or that have over one billion parameters in order to reduce computational costs. We then select from the top ten encoders as of August 2024 after applying these criteria, taking only the top encoder from a given family if several encoders from this family appear, and omitting encoders that were not compatible with the sentence-transformers library. The final encoders selected from MTEB were stella-en-400M-v5 (hug, a), gte-large-en-v1.5 (Zhang et al., 2024b) (Li et al., 2023), GIST-small-Embedding-v0 (Solatorio, 2024), ember-v1 (Nur & Aliyev, 2023), bge-large-en-v1.5 (Xiao et al., 2023), and mxbai-embed-large-v1 (Li & Li, 2023) (Lee et al., 2024). These encoders were tested against all-MiniLM-L12-v2 (hug, b) (the encoder used for CELLama) (Choi et al., 2024), OpenAI ada-002 (the encoder used for GenePT) (Chen & Zou, 2023) and OpenAI's newest text embedding model text-embedding-3-large (OpenAI, a).

To evaluate the cell type classification and clustering performance of these text encoders, we base our experimental design on the GenePT paper. For classification, we apply a 10-nearest neighbour classifier, classifying each cell in the test set of a given dataset based on the labels of its 10-nearest neighbours from the corresponding train set, measured using cosine similarity between cell embeddings. Accuracy, along with macro-weighted precision, recall and F1 score are then reported. To evaluate cell type clustering, for each encoder on each dataset, we apply k-means clustering on the cell embeddings, setting the number of clusters to match the number of cell types for the given dataset. We then compute the Adjusted Rand Index (ARI) and Adjusted Mutual Information (AMI) to evaluate the concordance between the resultant clusters and the true cell type labels.

We first compare the cell type classification and clustering performance of all encoders of potential interest on the Aorta dataset (Li et al., 2020). Results are presented in Table 5. We observe that ada-002 and all-MiniLM-L12-v2 were both outperformed by several of the encoders selected from MTEB on both cell type classification and clustering. Ember-V1 performed particularly well, outperforming both of these encoders by a wide margin on this dataset. We also observe that text-embedding-3-large performed substantially worse than ada-002, despite it being a newer text embedding model from OpenAI designed to improve performance (OpenAI, a).

Table 5: Zero-shot cell type classification and clustering performance of encoders on the Aorta dataset. Highest value for each metric in bold.

| Model | Accuracy | Precision | Recall | F1 | k-means ARI | k-means AMI |
|---|---|---|---|---|---|---|
| **ada-002** | 0.872 | 0.865 | 0.670 | 0.716 | 0.350 | 0.510 |
| **text-embedding-3-large** | 0.791 | 0.570 | 0.438 | 0.451 | 0.160 | 0.200 |
| **all-MiniLM-L12-v2** | 0.855 | 0.731 | 0.623 | 0.644 | 0.441 | 0.495 |
| **Ember-V1** | **0.906** | **0.910** | 0.800 | **0.841** | **0.535** | **0.597** |
| **gte-large-en-v1.5** | 0.894 | 0.903 | 0.746 | 0.791 | 0.442 | 0.529 |
| **mxbai-embed-large-v1** | 0.901 | 0.885 | 0.761 | 0.801 | 0.303 | 0.506 |
| **bge-large-en-v1.5** | 0.905 | 0.902 | 0.799 | 0.837 | 0.405 | 0.546 |
| **GIST-small-embedding-v0** | 0.871 | 0.859 | 0.696 | 0.735 | 0.342 | 0.482 |
| **stella_en_400m_v5** | 0.905 | 0.903 | **0.804** | 0.838 | 0.323 | 0.552 |

We then compare Ember-V1 against ada-002, allMiniLM-L12-V2, and scGPT on all the datasets collected to evaluate cell type classification performance in this study. This includes the MS (Schirmer et al., 2019), Artery (Alsaigh et al., 2022), Bones (Chou et al., 2020), Myeloid (Cheng et al., 2021), Pancreas(Luecken et al., 2022), and subsampled Tabula Sapiens (Consortium* et al., 2022) datasets. Results for classification and clustering performance are reported in Tables 6, 7, 8, 9, 10, 11, and 12. Overall, Ember-V1 outperforms allMiniLM-L12-V2 on every metric on every dataset. The improvement in performance over ada-002 is more modest, with improved clustering and classification performance on 5/7 datasets tested overall. However, ada-002 is not an open source model, which makes many interpretability methods challenging or impossible to use. Ember-

V1 also performs competitively with scGPT on clustering and classification, with better performance on 3/7 datasets.

Ultimately, we find that Ember-V1 outperforms previously used text embedding models in generating cell embeddings from cell sentences, motivating us to select this encoder for our main experiments.

Table 6: Zero-shot cell type classification and clustering performance of different encoders on the Aorta dataset.

| Metric | ada-002 | Ember-V1 | scGPT | all-MiniLM-L12-v2 |
|---|---|---|---|---|
| Accuracy | 0.872 | 0.906 | **0.960** | 0.855 |
| Precision | 0.865 | 0.910 | **0.958** | 0.731 |
| Recall | 0.670 | 0.800 | **0.942** | 0.623 |
| F1 | 0.716 | 0.841 | **0.949** | 0.644 |
| K-means ARI | 0.350 | **0.535** | 0.463 | 0.441 |
| K-means AMI | 0.510 | 0.597 | **0.637** | 0.495 |

Table 7: Zero-shot cell type classification and clustering performance of different encoders on the Myeloid dataset.

| Metric | ada-002 | Ember-V1 | scGPT | all-MiniLM-L12-v2 |
|---|---|---|---|---|
| Accuracy | 0.518 | 0.499 | **0.545** | 0.497 |
| Precision | **0.359** | 0.333 | 0.336 | 0.330 |
| Recall | 0.287 | 0.266 | **0.294** | 0.254 |
| F1 | **0.306** | 0.284 | **0.306** | 0.268 |
| K-means ARI | 0.297 | 0.283 | **0.414** | 0.241 |
| K-means AMI | 0.393 | 0.409 | **0.516** | 0.304 |

Table 8: Zero-shot cell type classification and clustering performance of different encoders on the MS dataset.

| Metric | ada-002 | Ember-V1 | scGPT | all-MiniLM-L12-v2 |
|---|---|---|---|---|
| Accuracy | 0.460 | 0.444 | **0.752** | 0.416 |
| Precision | 0.467 | 0.457 | **0.667** | 0.415 |
| Recall | 0.380 | 0.338 | **0.616** | 0.331 |
| F1 | 0.360 | 0.325 | **0.596** | 0.319 |
| K-means ARI | 0.247 | 0.185 | **0.292** | 0.168 |
| K-means AMI | 0.339 | 0.394 | **0.480** | 0.298 |

Table 9: Zero-shot cell type classification and clustering performance of different encoders on the Pancreas dataset.

| Metric | ada-002 | Ember-V1 | scGPT | all-MiniLM-L12-v2 |
|---|---|---|---|---|
| Accuracy | 0.983 | **0.984** | 0.784 | 0.978 |
| Precision | **0.805** | 0.796 | 0.594 | 0.770 |
| Recall | 0.742 | **0.767** | 0.550 | 0.667 |
| F1 | 0.769 | **0.778** | 0.545 | 0.699 |
| K-means ARI | 0.487 | **0.864** | 0.202 | 0.456 |
| K-means AMI | 0.740 | **0.827** | 0.411 | 0.666 |

Table 10: Zero-shot cell type classification and clustering performance of different encoders on the Bones dataset.

| Metric | ada-002 | Ember-V1 | scGPT | all-MiniLM-L12-v2 |
|--------|---------|----------|-------|-------------------|
| Accuracy | 0.353 | **0.427** | 0.326 | 0.311 |
| Precision | 0.372 | **0.423** | 0.358 | 0.345 |
| Recall | 0.495 | **0.550** | 0.494 | 0.473 |
| F1 | 0.272 | **0.322** | 0.244 | 0.249 |
| K-means ARI | 0.165 | **0.251** | 0.098 | 0.127 |
| K-means AMI | 0.282 | **0.357** | 0.199 | 0.221 |

Table 11: Zero-shot cell type classification and clustering performance of different encoders on the Artery dataset.

| Metric | ada-002 | Ember-V1 | scGPT | all-MiniLM-L12-v2 |
|--------|---------|----------|-------|-------------------|
| Accuracy | 0.916 | 0.924 | **0.949** | 0.890 |
| Precision | 0.874 | 0.885 | **0.920** | 0.809 |
| Recall | 0.819 | 0.823 | **0.894** | 0.798 |
| F1 | 0.839 | 0.847 | **0.904** | 0.799 |
| K-means ARI | 0.358 | 0.490 | **0.533** | 0.346 |
| K-means AMI | 0.564 | 0.671 | **0.704** | 0.524 |

Table 12: Zero-shot cell type classification and clustering performance of different encoders on the Tabula Sapiens dataset.

| Metric | ada-002 | Ember-V1 | scGPT | all-MiniLM-L12-v2 |
|--------|---------|----------|-------|-------------------|
| Accuracy | 0.682 | **0.703** | 0.690 | 0.682 |
| Precision | 0.262 | **0.282** | 0.278 | 0.260 |
| Recall | 0.252 | 0.265 | **0.277** | 0.236 |
| F1 | 0.231 | **0.250** | 0.246 | 0.219 |
| K-means ARI | 0.311 | **0.521** | 0.197 | 0.410 |
| K-means AMI | 0.633 | **0.700** | 0.565 | 0.609 |

## A.2 SCMPT - ADDITIONAL DETAILS AND RESULTS

scMPT is trained using the AdamW optimizer (Chollet et al., 2015) (Loshchilov, 2017) (Kingma, 2014), and an exponential decay learning rate scheduler. The initial learning rate, number of epochs, batch size, and decay rate are selected for each dataset through a grid search. This hyperparameter tuning was conducted for each dataset using a validation split derived from the train set. The dense layers which the encoders feed into each have an output dimension of 4096, and use the ReLU (Agarap, 2018) activation function. The output layer uses a softmax activation function. Results for scMPT were averaged across 5 random seeds, with mean and standard deviation reported as Mean (Standard Deviation).

For the individual encoders with MLP classification heads, we use the default architecture for the scikit-learn library's MLP implementation (Pedregosa et al., 2011), and leave the text encoder frozen during training to reduce computational cost.

Table 13: scMPT cell type classification performance vs unimodal models - Aorta dataset.

| Model | Accuracy | Precision | Recall | F1 |
|---|---|---|---|---|
| scMPT | 0.971 (0.0008) | 0.967 (0.0019) | 0.954 (0.0013) | 0.960 (0.0012) |
| scGPT + MLP | 0.968 | 0.960 | 0.949 | 0.954 |
| Ember-V1 + MLP | 0.940 | 0.923 | 0.869 | 0.889 |

Table 14: scMPT cell type classification performance vs unimodal models - Artery dataset.

| Model | Accuracy | Precision | Recall | F1 |
|---|---|---|---|---|
| scMPT | 0.962 (0.00057) | 0.935 (0.00033) | 0.928 (0.0019) | 0.931 (0.0011) |
| scGPT + MLP | 0.961 | 0.932 | 0.926 | 0.929 |
| Ember-V1 + MLP | 0.949 | 0.910 | 0.899 | 0.903 |

Table 15: scMPT cell type classification performance vs unimodal models - Bones dataset.

| Model | Accuracy | Precision | Recall | F1 |
|---|---|---|---|---|
| scMPT | 0.684 (0.031) | 0.549 (0.0126) | 0.691 (0.0094) | 0.554 (0.0217) |
| Ember-V1 + MLP | 0.674 | 0.526 | 0.686 | 0.541 |
| scGPT + MLP | 0.630 | 0.509 | 0.657 | 0.498 |

Table 16: scMPT cell type classification performance vs unimodal models - Tsapeins dataset.

| Model | Accuracy | Precision | Recall | F1 |
|---|---|---|---|---|
| scMPT | 0.764 (0.0033) | 0.349 (0.0177) | 0.297 (0.0094) | 0.290 (0.0128) |
| scGPT + MLP | 0.736 | 0.304 | 0.274 | 0.261 |
| Ember-V1 + MLP | 0.748 | 0.291 | 0.236 | 0.236 |

Table 17: scMPT cell type classification performance vs unimodal models - Myeloid dataset.

| Model | Accuracy | Precision | Recall | F1 |
|---|---|---|---|---|
| scMPT | 0.664 (0.0026) | 0.387 (0.009) | 0.364 (0.0086) | 0.3694 (0.0087) |
| scGPT (fine-tuned) | 0.642 | 0.366 | 0.347 | 0.346 |
| scGPT + MLP | 0.622 | 0.380 | 0.349 | 0.354 |
| scGPT (from-scratch) | 0.606 | 0.304 | 0.339 | 0.309 |
| Ember-V1 + MLP | 0.601 | 0.352 | 0.314 | 0.325 |

Table 18: scMPT cell type classification performance vs unimodal models - MS dataset.

| Model | Accuracy | Precision | Recall | F1 |
|---|---|---|---|---|
| scGPT + MLP | 0.845 | 0.769 | 0.735 | 0.726 |
| scMPT | 0.837 (0.00089) | 0.733 (0.0039) | 0.718 (0.0024) | 0.704 (0.003) |
| scGPT (fine-tuned) | 0.856 | 0.729 | 0.720 | 0.703 |
| scGPT (from-scratch) | 0.798 | 0.660 | 0.623 | 0.600 |
| Ember-V1 + MLP | 0.687 | 0.597 | 0.582 | 0.568 |

Figures 2 and 3 below summarize scMPT cell type classification performance compared to scGPT; both for the setting where an MLP is trained on top of scGPT, and where k-nearest neighbours is used for zero-shot cell type classification.

Figure 2: Comparison between cell type classification accuracy of scMPT and scGPT on all datasets used in this study to evaluate performance on this task. scMPT outperforms scGPT on most datasets tested, with particularly significant improvements on the Pancreas and Bones datasets.

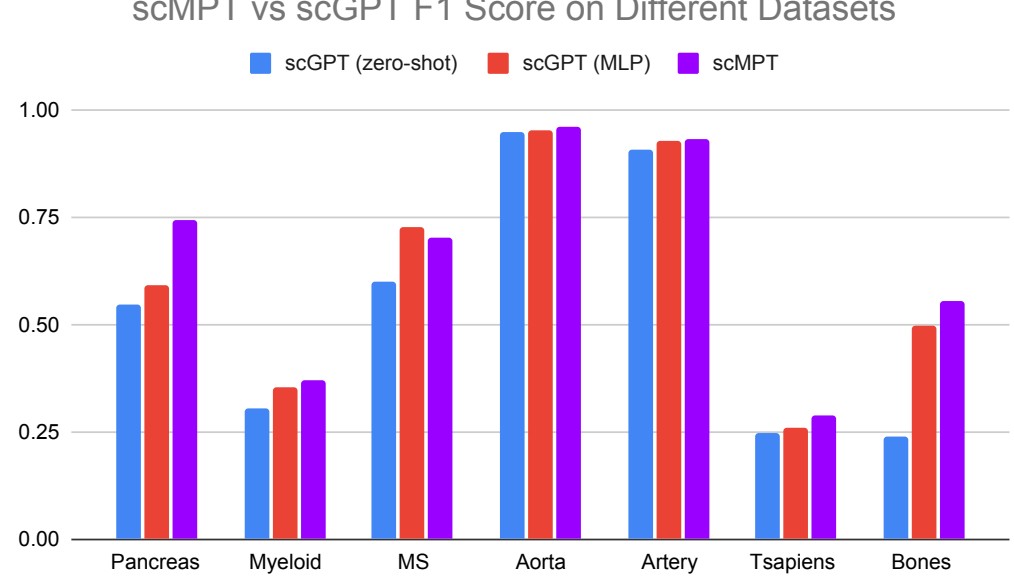

Figure 3: Comparison between cell type classification F1 score of scMPT and scGPT on all datasets used in this study to evaluate performance on this task. scMPT outperforms scGPT on most datasets tested, with particularly significant improvements on the Pancreas and Bones datasets.

We also test the performance of scMPT on disease (specifically, aneurysm) phenotype prediction using the Aorta dataset. We use the same train and test split as the one used to evaluate cell type classification.

Table 19: scMPT disease phenotype prediction performance vs unimodal models - Aorta dataset.

| Model | Accuracy | Precision | Recall | F1 |
|-------|----------|-----------|--------|-----|
| **scMPT** | 0.894 (0.00197) | 0.840 (0.00434) | 0.822 (0.00138) | 0.831 (0.00216) |
| **Ember-V1 + MLP** | 0.863 | 0.809 | 0.790 | 0.799 |
| **scGPT + MLP** | 0.862 | 0.810 | 0.786 | 0.797 |

We further evaluate disease phenotype prediction performance with the Cardiomyocyte dataset which was originally used, alongside the Aorta dataset, to evaluate disease phenotype prediction performance in the GenePT paper (Chaffin et al., 2022) (Chen & Zou, 2023).

Table 20: scMPT disease phenotype prediction performance vs unimodal models - Cardiomyocyte dataset.

| Model | Accuracy | Precision | Recall | F1 |
|-------|----------|-----------|--------|-----|
| **scMPT** | 0.953 (0.0000) | 0.952 (0.0016) | 0.951 (0.0015) | 0.951 (0.0000) |
| **scGPT + MLP** | 0.927 | 0.927 | 0.922 | 0.924 |
| **Ember-V1 + MLP** | 0.926 | 0.924 | 0.923 | 0.923 |

Finally, to further investigate the generalizability of the scMPT framework, we experiment with substituting scGPT for the Geneformer (Theodoris et al., 2023) single-cell foundation model. We refer to this model in the tables below as scMPT-gf.

Table 21: scMPT cell type classification performance vs unimodal models (scGPT replaced with Geneformer) - Aorta dataset.

| Model | Accuracy | Precision | Recall | F1 |
|---|---|---|---|---|
| **scMPT-gf** | 0.959 (0.0008) | 0.948 (0.0008) | 0.928 (0.0019) | 0.937 (0.0011) |
| **geneformer + MLP** | 0.950 | 0.922 | 0.919 | 0.920 |
| **Ember-V1 + MLP** | 0.940 | 0.923 | 0.869 | 0.889 |

Table 22: scMPT disease phenotype prediction performance vs unimodal models (scGPT replaced with Geneformer) - Aorta dataset.

| Model | Accuracy | Precision | Recall | F1 |
|---|---|---|---|---|
| **scMPT-gf** | 0.906 (0.0024) | 0.859 (0.0064) | 0.844 (0.0090) | 0.851 (0.0030) |
| **geneformer + MLP** | 0.898 | 0.866 | 0.823 | 0.840 |
| **Ember-V1 + MLP** | 0.863 | 0.809 | 0.790 | 0.799 |

### A.3 GENERATIVE LLM PROMPTS, AND ADDITIONAL RESULTS

#### A.3.1 EVALUATING THE PERFORMANCE OF GENERATIVE LLMS FOR ZERO-SHOT CELL TYPE CLASSIFICATION

To better understand the impact of reasoning on generative LLM performance in single-cell analysis, we compare the performance of standard and reasoning models on zero-shot cell type classification, without combining these models with scGPT. To classify individual cells using generative LLMs, we pass in the cell sentence for the cell we want to classify, and instruct the model to output the most likely cell type given a list of all cell types from the corresponding train set. The prompt used is shown in Figure 4. Providing this list of cell types to narrow the LLM's output space, a change from previous work, facilitates a more fair comparison of classification performance with scGPT, evaluated in this section for reference, since k-nearest neighbours will also only output labels present in the train set. To limit costs, we focused on the three datasets used to evaluate cell-type annotation in the scGPT paper, namely the Myeloid (Cheng et al., 2021), Pancreas (Luecken et al., 2022), and Multiple Sclerosis (MS) (Schirmer et al., 2019) datasets. For each dataset we evaluated cell type classification performance on a subset of 100 randomly selected cells from the test set, reporting accuracy. Note that the same 100 cells from each dataset are used to evaluate all models. These are also the same 100 cells from each dataset used in Section 4.2.

After evaluating the performance of GPT-4o using this setup, we compare it to o3-mini to investigate the impact of switching to a model specifically trained for improved reasoning. We also investigate the performance of open source models in this setting, namely DeepSeek-V3, and DeepSeek-R1 in order to better understand the impact of switching to a reasoning model. Finally, we report the performance of scGPT as a reference. We report classification accuracy for all models in Table 23. We report mean and standard deviation across 3 runs for each dataset.

Table 23: Cell type classification accuracy of generative LLMs on different datasets when provided with a list of possible labels, reported as Mean (Standard Deviation). Models with reasoning specific fine tuning underlined. scGPT included for reference.

| Model | Pancreas Data | Myeloid Data | MS Data |
|---|---|---|---|
| scGPT | 0.78 | 0.51 | 0.75 |
| DeepSeek-V3 | 0.953 (0.012) | 0.327 (0.006) | 0.420 (0.000) |
| DeepSeek-R1 | 0.970 (0.010) | 0.307 (0.023) | 0.450 (0.020) |
| gpt-4o | 0.970 (0.010) | 0.280 (0.030) | 0.350 (0.000) |
| o3-mini | 0.980 (0.010) | 0.393 (0.015) | 0.473 (0.012) |

In general, the performance of generative LLMs was reasonably strong. While these models were outperformed overall by scGPT, their performance was comparable to this specialized foundation model, and notably much better on the Pancreas dataset. This is a dramatic improvement compared to previously reported results, which found that GPT-4 was unable to classify any cells correctly in this setting of classifying individual cells (Liu et al., 2023). Finally, we find that models that were specifically trained for reasoning exhibited stronger performance than the others tested, with o3-mini performing particularly well.

### A.3.2 ADDITIONAL INFORMATION AND PROMPTS

For the experiments involving cell type classification, and using generative LLMs to complement scGPT, the specific versions of the generative LLMs used are *gpt-4o-2024-11-20*, *o3-mini-2025-01-31*, and the original versions of DeepSeek-R1 and DeepSeek-V3 hosted by Fireworks AI (https://fireworks.ai/models/fireworks/deepseek-r1) (https://fireworks.ai/models/fireworks/deepseek-v3) (AI, 1/20/2025)(AI, 12/30/2024). For DeepSeek-R1, a temperature of 0.6 was used, and for DeepSeek-V3, a temperature of 0 was used based on DeepSeek's recommendations (DeepSeek, b) (DeepSeek, a). Default parameters were used for OpenAI's models. Structured outputs were used for cell type classification, forcing models to only output a predicted cell type label for a given cell and nothing else. This facilitated a more fair evaluation of LLMs with different instruction following capabilities, while enabling us to simplify prompt structure, and use the same prompts across models and datasets.

For the LLM + scGPT Prompt in Figure 5, we instruct the model to select from the 3 most likely cell types predicted by scGPT based on the large gap in scGPT's top-3 and top-1 performance, as shown in Table 24. The 3 most likely cell types are the 3 cell types most represented in the cell of interest's 10 nearest neighbours.

> **LLM Prompt**
>
> ```
> Given the cell described. Output ONLY the
> most likely cell type from the following
> list: + [cell types from train set]
> + """{cell sentence}"""
> ```

Figure 4: Prompt used to classify cell type using generative LLMs.

> **LLM + scGPT Prompt**
>
> ```
> Given the cell described. Output ONLY the
> most likely cell type from the following
> list:  + [3 most likely cell types
> according to scGPT] + If you are not
> certain about what cell type this is,
> conclude that it is the first option in
> the list
> + """[cell sentence]"""
> ```

Figure 5: Prompt used to enable generative LLMs to complement scGPT.

Table 24: scGPT Top-1 and Top-3 Accuracy on the Pancreas, Myeloid, and MS datasets.

| Top-K | Pancreas | Myeloid | MS |
|-------|----------|---------|------|
| Top-1 | 0.78 | 0.51 | 0.75 |
| Top-3 | 0.93 | 0.75 | 0.89 |

### A.3.3 ADDITIONAL RESULTS - GENERATIVE LLM KNOWLEDGE OF MARKER GENES

To evaluate generative LLMs' knowledge of marker genes, we evaluate whether these models are able to match top marker genes from PanglaoDB (Franzén et al., 2019) to other markers for the same cell type. Specifically, for each of the Pancreas and Aorta datasets, we construct a list of the top 5 marker genes from PanglaoDB for each cell type. We then remove the top gene from each list, and see if the LLM can match each of these top genes with the correct list of remaining 4 markers. We find that performance on this task improves with scale in terms of number of model parameters, and that DeepSeek-V3 and GPT-4o perform well, indicating that they are able to effectively leverage knowledge of marker genes. We focus on DeepSeek-V3 and GPT-4o as our goal is to assess the knowledge of models in this experiment rather than reasoning capabilities. The Llama family of models is studied and compared to GPT-4o because it consists of different size models that generally demonstrate strong performance for their scale (Dubey et al., 2024). We use the implementations of these models hosted by Replicate (Replicate). All API calls for this experiment use a temperature of 0.

Table 25: Number of Top Marker Genes Matched Correctly by Different Generative LLMs for the Pancreas Dataset.

| Model | Genes Correctly Matched |
|-------|-------------------------|
| Llama 3 8B | 3/10 |
| Llama 3 70B | 4/10 |
| Llama 3.1 405B | 6/10 |
| GPT-4o | 7/10 |
| DeepSeek-V3 (671B - 37B Active) | 8/10 |

Table 26: Number of Top Marker Genes Matched Correctly by Different Generative LLMs for the Aorta Dataset.

| Model | Genes Correctly Matched |
|-------|-------------------------|
| Llama 3 8B | 2/6 |
| Llama 3 70B | 4/6 |
| Llama 3.1 405B | 4/6 |
| GPT-4o | 4/6 |
| DeepSeek-V3 (671B - 37B Active) | 5/6 |

## A.4 ADDITIONAL RESULTS FOR ABLATION STUDY

Below are results for all ablations for ada-002 on the Aorta dataset.

Table 27: Ablation test results - Aorta Data (ada-002 - zero-shot / KNN).

| Metric | Original sentence length | Baseline (Ember-V1 Context Length) | Gene Name Ablation | Order Ablation (All Genes) | Order Ablation (In Context) | Order Ablation (Top 10% In Context) | Gene Name + Order Ablation (In Context) |
|---|---|---|---|---|---|---|---|
| Accuracy | 0.872 | 0.889 | 0.838 | 0.337 | 0.697 | 0.808 | 0.535 |
| Precision | 0.865 | 0.916 | 0.883 | 0.084 | 0.483 | 0.842 | 0.253 |
| Recall | 0.670 | 0.765 | 0.567 | 0.091 | 0.303 | 0.509 | 0.177 |
| F1 | 0.716 | 0.804 | 0.615 | 0.077 | 0.311 | 0.560 | 0.171 |
| k-means ARI | 0.350 | 0.363 | 0.477 | -0.001 | 0.203 | 0.336 | 0.010 |
| k-means AMI | 0.510 | 0.538 | 0.451 | 0.000 | 0.219 | 0.419 | 0.005 |

Next, we present Ember-V1 ablation results on the two datasets where it substantially outperformed scGPT; the Pancreas and Bones datasets.

Table 28: Ablation test results – Pancreas Data (Ember-V1 – zero-shot / KNN).

| Metric | No Ablations / Baseline | Gene Name Ablation | Order Ablation (All Genes) | Order Ablation (In Context) | Order Ablation (Top 10% In Context) | Gene Name + Order Ablation (In Context) | Gene Name Per-Instance Ablation |
|---|---|---|---|---|---|---|---|
| Accuracy | 0.984 | 0.976 | 0.308 | 0.871 | 0.925 | 0.631 | 0.342 |
| Precision | 0.796 | 0.802 | 0.084 | 0.672 | 0.790 | 0.338 | 0.092 |
| Recall | 0.767 | 0.748 | 0.086 | 0.564 | 0.657 | 0.301 | 0.092 |
| F1 | 0.778 | 0.771 | 0.082 | 0.579 | 0.685 | 0.294 | 0.086 |
| k-means ARI | 0.864 | 0.881 | -0.003 | 0.549 | 0.538 | 0.035 | 0.000 |
| k-means AMI | 0.827 | 0.865 | -0.001 | 0.553 | 0.625 | 0.100 | 0.001 |

Table 29: Ablation test results – Bones Data (Ember-V1 – zero-shot / KNN).

| Metric | No Ablations / Baseline | Gene Name Ablation | Order Ablation (All Genes) | Order Ablation (In Context) | Order Ablation (Top 10% In Context) | Gene Name + Order Ablation (In Context) | Gene Name Per-Instance Ablation |
|---|---|---|---|---|---|---|---|
| Accuracy | 0.427 | 0.282 | 0.031 | 0.199 | 0.344 | 0.049 | 0.030 |
| Precision | 0.423 | 0.424 | 0.031 | 0.284 | 0.393 | 0.180 | 0.074 |
| Recall | 0.550 | 0.428 | 0.140 | 0.345 | 0.493 | 0.206 | 0.149 |
| F1 | 0.322 | 0.228 | 0.014 | 0.161 | 0.262 | 0.039 | 0.016 |
| k-means ARI | 0.251 | 0.101 | 0.001 | 0.136 | 0.209 | 0.010 | 0.001 |
| k-means AMI | 0.357 | 0.165 | 0.000 | 0.199 | 0.264 | 0.016 | -0.001 |

Based on the above results, we then explore the effect of using shorter cell sentences with Ember-V1. We first conduct this experiment on the Aorta dataset, followed by the Bones and Pancreas dataset, using ada-002 and scGPT as a baseline. Note that "% of Cell Sentence" refers to "% of In-Context Cell Sentence".

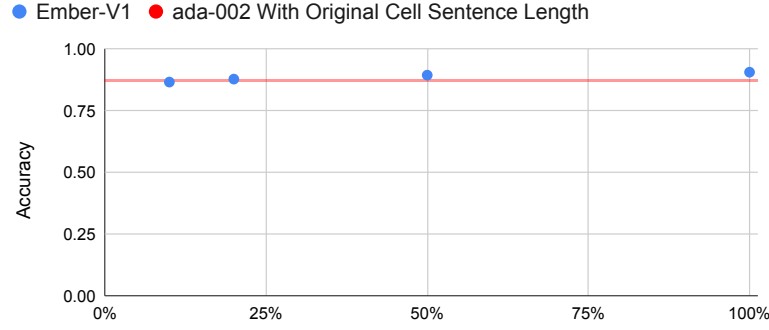

Figure 6: Accuracy (Top), and F1 Score (Bottom) of Ember-V1 on Aorta Dataset with different cell sentence lengths. Performance of ada-002 using all genes included as a baseline.

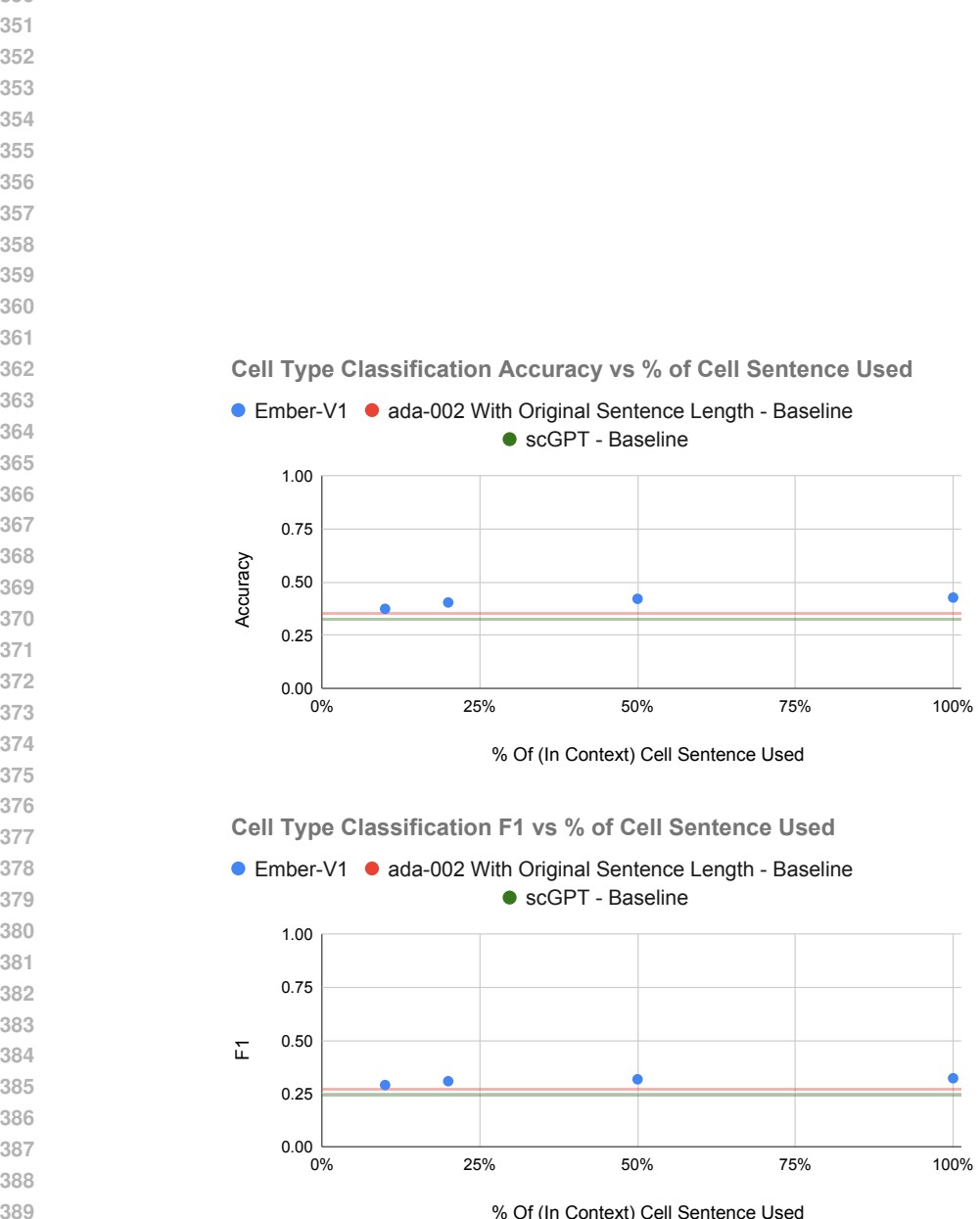

Figure 7: Accuracy (Top), and F1 Score (Bottom) of Ember-V1 on Bones Dataset with different cell sentence lengths. Performance of ada-002 and scGPT using all genes included as a baseline.

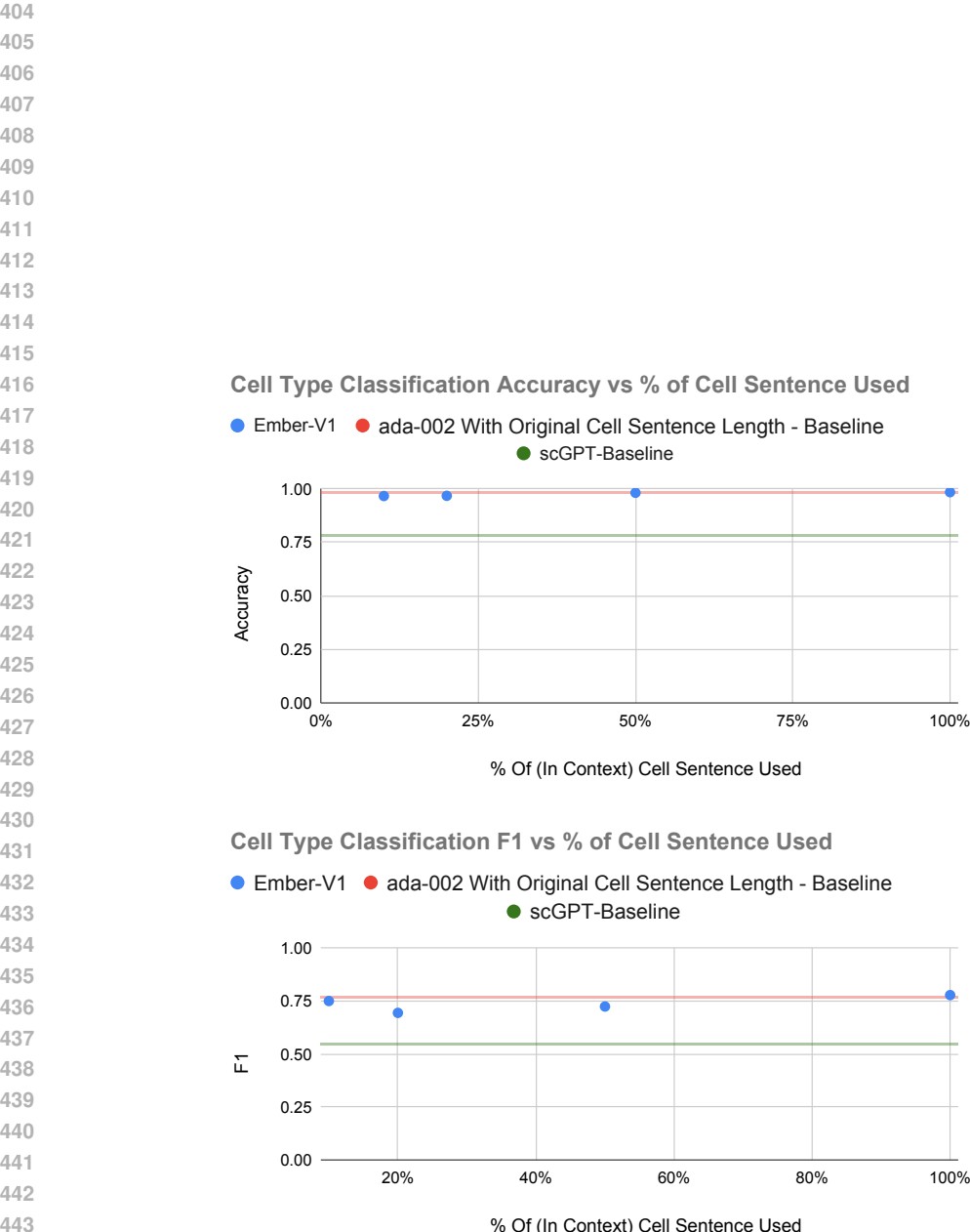

Figure 8: Accuracy (Top), and F1 Score (Bottom) of Ember-V1 on Pancreas Dataset with different cell sentence lengths. Performance of ada-002 and scGPT using all genes included as a baseline.

Next, we present results for the gene name ablation test (where gene names are replaced with truncated SHA-256 hashes) for Ember-V1 on other datasets, reporting classification performance before and after this ablation.

Table 30: Ember-V1 cell type classification performance before and after gene name ablation on datasets used for evaluation in the scGPT paper. (Cui et al., 2024).

| Metrics | Pancreas Data | | MS Data | | Myeloid Data | |
|---|---|---|---|---|---|---|
| | Ember-V1 Ablated | Ember-V1 | Ember-V1 Ablated | Ember-V1 | Ember-V1 Ablated | Ember-V1 |
| accuracy | 0.9755 | 0.984 | 0.364 | 0.445 | 0.483 | 0.499 |
| precision | 0.803 | 0.796 | 0.341 | 0.456 | 0.317 | 0.333 |
| recall | 0.748 | 0.767 | 0.283 | 0.338 | 0.244 | 0.266 |
| F1 | 0.771 | 0.778 | 0.273 | 0.325 | 0.261 | 0.284 |

Table 31: Ember-V1 cell type classification performance before and after gene name ablation on other datasets.

| Metrics | Tabula Sapiens Data | | Artery Data | | Bones Data | |
|---|---|---|---|---|---|---|
| | Ember-V1 Ablated | Ember-V1 | Ember-V1 Ablated | Ember-V1 | Ember-V1 Ablated | Ember-V1 |
| accuracy | 0.534 | 0.702 | 0.877 | 0.924 | 0.282 | 0.427 |
| precision | 0.217 | 0.288 | 0.805 | 0.885 | 0.424 | 0.421 |
| recall | 0.181 | 0.265 | 0.743 | 0.823 | 0.429 | 0.551 |
| F1 | 0.162 | 0.252 | 0.767 | 0.847 | 0.228 | 0.321 |

A.5 ADDITIONAL RESULTS FOR INTERPRETABILITY TESTS

LIME and integrated gradients results for the Aorta Dataset are provided below.

Table 32: Marker genes in top 10 gene attributions for Ember-V1 encoder + MLP with different interpretability methods - Aorta dataset.
*(Markers that were highlighted by both methods in bold)*

| Cell Type | LIME | Integrated Gradients |
|---|---|---|
| NK | **KLRB1**, **NKG7**, **XCL1**, XCL2, KLRD1 | **KLRB1**, **XCL1**, **NKG7** |
| T Cell | S100A4, TNFAIP3 | |
| B Cell | **HLA-DRA**, CD74, IGKC | **HLA-DRA** |
| Fibroblast | **MGP**, CTGF, LUM, COL1A2, DCN | IGFBP6, **MGP** |
| Mast Cell | **TPSAB1**, TPSB2, RGS1, KIT | **TPSAB1** |
| Plasma Cell | **IGHG1**, **IGKC**, **IGHG3**, IGHA1, **IGHG4**, IGLC2, JCHAIN | **IGHG3**, **IGKC**, **IGHG4**, **IGHG1** |

Results for our analysis to get a more global understanding of Ember-V1 and its knowledge of marker genes are provided in Table 33.

Table 33: Average cosine similarity between Ember-V1 embeddings of top marker gene names, and markers of the same cell type (Intra-Similarity), and different cell types (Inter-Similarity).

| Dataset | Intra-Similarity | Inter-Similarity |
|---------|------------------|------------------|
| Pancreas | **0.644** | 0.623 |
| Aorta | **0.667** | 0.653 |

## A.6 ADDITIONAL EXPERIMENTAL DETAILS

### A.6.1 STANDARD DEVIATION CALCULATION

Standard deviations reported are sample standard deviations, calculated using the following formula:

$$s = \sqrt{\frac{1}{n-1} \sum_{i=1}^{n} (x_i - \bar{x})^2}$$

Where $n$ represents the total number of samples (or data points), each $x_i$ is an individual sample value, and $\bar{x}$ denotes the sample mean, calculated as $\bar{x} = \frac{1}{n} \sum_{i=1}^{n} x_i$.

## A.7 LLM USAGE DECLARATION

LLMs were used to help edit and polish writing. They were also used to assist in writing code.

