# OpenReview forum: "Bridging Gene Expression and Text: LLMs Can Complement Single-Cell Foundation Models"
_ICLR.cc/2026/Conference — Submitted to ICLR 2026_

### Official Review · Reviewer_615z · 2025-10-28

**Soundness:** 1
**Presentation:** 3
**Contribution:** 1
**Rating:** 2
**Confidence:** 4

**Summary:**

This paper explores methods for bridging the gap between text-based Large Language Models (LLMs) and specialized single-cell foundation models trained on raw  single-cell data. Single-cell (gene expression) data is a high-dimensional vector of gene activity counts for a single cell that is usually very sparse (only a few genes are active/expressed in each cell at a time). The authors' core methodology is to convert this vector into a "cell sentence," which is a text string of gene names, ranked by their expression level. The main intuition/idea behind this conversion is to tap into the vast textual knowledge of LLMs, namely from curated datasets about gene markers for a specific biological class.  Namely, an LLM trained on medical literature would likely learn the association between the word "insulin" and the phrase "Beta cell."

The paper has two parts: First, it presents an interpretability study to investigate what biological insights LLMs learn from these "cell sentences." The authors use  standard interpretability techniques (LIME and Integrated Gradients) to uncover what "marker genes" where used for a prediction. They report that the genes the model found "important" were, in fact, biologically-known marker genes, confirming one of the parts of the hypothesis.  Second, the paper introduces scMPT, a simple fusion model. This model takes the frozen embeddings from a single-cell model (scGPT) and a text-encoder LLM (Ember-V1), concatenates them, and feeds them into a small, trainable MLP. The authors claim this fusion model achieves strong performance, even suggesting it is "competitive with full fine tunes of scGPT" on tasks like cell-type classification. The claims for this second part are highly questionable as discussed in weaknesses section.

**Strengths:**

The paper's only methodologically sound contribution is the interpretability analysis in Section 4.1. This section is well-executed and validates that the authors' model learns to identify key biological features. However, the novelty of this analysis is limited, as the core finding—that a model learns marker-gene associations for cell typing, validated against the PanglaoDB database—was previously demonstrated by the scBERT paper (Yang et al., 2022) . The paper's novel contribution is therefore a methodological validation in successfully replicating this concept on a different class of model (a general-purpose text encoder, not a specialized biological model). Therfore, this is a good but incremental validation rather than a novel conceptual discovery.

**Weaknesses:**

The paper has several critical flaws in methodology which leaves its central claims unsubstantiated.

- The paper's headline claim is that its scMPT model is "even competitive with full fine tunes of scGPT." The key evidence is in Table 3, where scMPT (F1=0.745) appears to beat the "scGPT (full fine-tune - reported)" baseline (F1=0.718) on the Pancreas dataset, which is a highly surprising fact. However, the scMPT's score of (F1=0.745) was tested on an **intra-study** benchmark, as elaborated here:

> We focus our experiments on the datasets that were used to evaluate the cell type classification and clustering performance of GenePT (Chen & Zou, 2023)... For each dataset, we use the **same train/test split** as each of these respective works ...

The cited GenePT and other cited work use a randomized intra-study (eg. they note "For the Aorta dataset, we used a random 80%/20% train/test split. ").  In contrast, the scGPT baseline (F1=0.718), was generated on a much harder, **cross-study** benchmark, elaborated here:

> The human pancreas dataset contains data from five scRNA-seq studies... The five datasets were **split into reference and query sets by data sources**. The reference set consists of data from **two data sources**, and the query set contains the **other three**.

Therefore, the authors are comparing their high score on a much simpler intra-study split task to a baseline's score on a hard generalization task, which invalidates their primary claim of SOTA-competitiveness.


-  Flawed claim of minimal loss under "cell sentence" conversion: The paper builds upon on converting numerical expression data into "cell sentences, and and cites Levine et al. (2023) to claim  "minimal information loss" under such a conversion.  However, in a more careful examination of the cited paper (Levine et al. 2023), the  evidence shows that a linear model that attempts to reconstruct the expression values from the ranks, achieving an $R^2$ score of 0.815 (Figure 3 in Levine et al.). This means that nearly **20%** of the variance in the original data is lost during the conversion to a "cell sentence." Representing a 20% loss of information as a "minimal loss" without citing an exact figure is a misrepresentation of what the literature shows.

- It also fails to cite and discuss CellPLM (Wen et al. 2023), a highly relevant related work that directly critiques the paper's "cell as sentence" methodology, arguing it misses crucial cell-cell relationships.
- Missing baseline: The paper's empirical evaluation fails to compare against scBERT (Yang et al. 2022), another prominent transformer-based foundation model for this exact task. (this paper is cited but not used as a baseline)

**Questions:**

Please answer the issues raised above. In particular, I look forward to hear author's responses and clarifications regarding the train/test split, which is my most important concern.

---

> ### Author Response · Authors · 2025-11-24
> **(1/2)**
>
> We thank the reviewer for their comments. We address the reviewer’s concerns below, and provide clarifications about what we believe are misunderstandings related to our methodology.
>
> > The cited GenePT and other cited work use a randomized intra-study (eg. they note "For the Aorta dataset, we used a random 80%/20% train/test split. "). In contrast, the scGPT baseline (F1=0.718), was generated on a much harder, cross-study benchmark, elaborated here: “The human pancreas dataset contains data from five scRNA-seq studies... The five datasets were split into reference and query sets by data sources. The reference set consists of data from two data sources, and the query set contains the other three.” Therefore, the authors are comparing their high score on a much simpler intra-study split task to a baseline's score on a hard generalization task, which invalidates their primary claim of SOTA-competitiveness.
>
> We use the same split as scGPT (which GenePT does as well). The 80/20 train/test split mentioned in the GenePT paper specifically refers to the Aorta dataset, not the Pancreas dataset. See the GenePT reporting summary (https://www.nature.com/articles/s41551-024-01284-6), and official Github repository: (https://github.com/yiqunchen/GenePT). Which point to https://drive.google.com/drive/folders/1s9XjcSiPC-FYV3VeHrEa7SeZetrthQVV?usp=drive_link, which is the link for the Pancreas dataset provided in the “Datasets for cell type annotation” section of scGPT’s official Github repository (https://github.com/bowang-lab/scGPT/tree/main/data). We use the datasets from this link labeled as train (demo_train) and test (demo_test) sets as our train and test sets.
>
> > Flawed claim of minimal loss under "cell sentence" conversion: The paper builds upon on converting numerical expression data into "cell sentences, and and cites Levine et al. (2023) to claim "minimal information loss" under such a conversion. However, in a more careful examination of the cited paper (Levine et al. 2023), the evidence shows that a linear model that attempts to reconstruct the expression values from the ranks, achieving an
>  score of 0.815 (Figure 3 in Levine et al.). This means that nearly 20% of the variance in the original data is lost during the conversion to a "cell sentence." Representing a 20% loss of information as a "minimal loss" without citing an exact figure is a misrepresentation of what the literature shows.
>
> We did not misrepresent the findings of this work as the phrasing “minimal loss” comes directly from it. See the introduction from Levine et al. (2023) (https://www.biorxiv.org/content/10.1101/2023.09.11.557287v1.full.pdf+html), where it is stated that:
>
> “C2S transforms each cell’s gene expression profile into a plaintext sequence of gene names (also known as gene symbols) ordered by expression level (Figures 1 and 2). Importantly, we show this rank transformation can be reverted with minimal loss of information.”
>
> > the authors' model learns to identify key biological features. However, the novelty of this analysis is limited, as the core finding—that a model learns marker-gene associations for cell typing, validated against the PanglaoDB database—was previously demonstrated by the scBERT paper (Yang et al., 2022) .
>
> We believe this misunderstands the central finding of this section, which is that LLMs leverage knowledge of marker genes from their parametric knowledge (ie. from training on large amounts of general text data), not that they obtain a knowledge of marker genes through training on single-cell data like the scBERT single-cell foundation model. Note that Ember-V1 is frozen in our analysis. This analysis is novel since, despite the application of LLMs to single-cell transcriptomics receiving significant recent interest because of the knowledge they obtain from training on general text data, the question of what knowledge of biology these models actually leverage in this setting is underexplored.

---

> ### Author Response · Authors · 2025-11-24
> **(2/2)**
>
> > It also fails to cite and discuss CellPLM (Wen et al. 2023), a highly relevant related work that directly critiques the paper's "cell as sentence" methodology, arguing it misses crucial cell-cell relationships.
>
> This work does not appear to be “highly” relevant, as it mainly provides a critique of how single-cell foundation models like scGPT are trained. It doesn’t discuss combining and exploring synergies between representations from LLMs and single-cell foundation models, which is the focus of our work. However, we agree with the reviewer that it provides an argument for why cell sentences are potentially not the end game for textual representations, which we discuss briefly in our future work section. Expanding this discussion with a citation to CellPLM could therefore be valuable, and we will revise accordingly. We thus sincerely thank the reviewer for this valuable feedback.
>
>
> > Missing baseline: The paper's empirical evaluation fails to compare against scBERT (Yang et al. 2022), another prominent transformer-based foundation model for this exact task. (this paper is cited but not used as a baseline)
>
> We agree with the reviewer that experimenting with single-cell foundation models other than scGPT is important, and note that we experimented with Geneformer as an alternate single-cell foundation model (as discussed in Section 4.2). We do not believe scBERT is more valuable to compare to than Geneformer.
>
> We hope our rebuttal addresses the reviewer’s concerns and clarifies what we believe to be misunderstandings of our methodology. If the reviewer is satisfied with our response, we would appreciate if they would consider revising their assessment of our work. We are also happy to engage in further discussion.

---

> > ### Author Response · Authors · 2025-11-28
> >
> > We would like to thank the reviewer 615z once again for their comments, and for their time spent reviewing our work. Since the end of the discussion period is fast approaching, we would like to know if the reviewer has any remaining questions or concerns so that we can have a chance to address them. We would like to note that in our rebuttal we addressed each of the reviewer's concerns, including clarifying our train/test split methodology, and why it is correct and appropriate. If our rebuttal is satisfactory in addressing the reviewer's concerns, we would appreciate it if the reviewer would consider revising their assessment of our paper. We thank the reviewer once again for their time.

---

### Official Review · Reviewer_F1vo · 2025-10-31

**Soundness:** 2
**Presentation:** 2
**Contribution:** 2
**Rating:** 4
**Confidence:** 3

**Summary:**

The authors pose the question on what LLMs learn from “cell sentences” and whether they complement single-cell FMs like scGPT. The paper finds (via IG+LIME and ablations) that LLMs lean on marker genes plus simple expression-rank patterns, then proposes scMPT, a light fusion that combines frozen Ember-V1 text embeddings with frozen scGPT features. scMPT typically matches or beats each component; a second scheme uses reasoning LLMs (e.g., o3-mini) to choose among scGPT’s top-3 labels and improves accuracy on several sets.

**Strengths:**

- The question is clearly formulated and the evidence are solidly given with interpretable analyses using integrated gradients and LIME against PanglaoDB markers.
- Given the experimental results, scMPT (frozen encoders + tiny MLPs) yields consistent gains; even reasoning-LLM reranking over scGPT’s top-3 helps.
- Some disease-phenotype results suggest benefits aren’t limited to cell-type classification.
- Hashing names and shuffling ranks show performance diffs on top ~10% in-context genes and their order, supporting the “simple pattern” claim.

**Weaknesses:**

- It seems that cell-sentence bias remains, as findings show reliance on high-rank genes/order; it’s unclear how robust this is to batch shifts.
- Regarding the claims about the knowledge of marker genes, we observe only some correlational evidence. In order to assure such claim, a stronger causal probe would be necessary, for example through counterfactual token tests.
- I find the eval for generative LLMs limited and biased: The head-to-head with GPT-4o / o3-mini uses only 100 test cells per dataset and constrains the label space to a provided list. That yields to sample-inefficient and risks optimistic estimates (especially for few, separable labels).
- If I follow correctly the methodology, IG and LIME are applied to an MLP on frozen Ember-V1 embeddings, not to the end-to-end LLM or the generative-LLM setup. So attributions correspond the MLP+Ember composite, not to scMPT nor the reranking LLMs.
- My main concern is that the ablations might hint some shortcut learning: Hashing names wipes performance, which might mean heavy reliance on lexical identity rather than biological semantics. These patterns look like lexical+rank shortcuts, not robust biology. The authors should evaluate under batch shift and low-depth cells.

**Questions:**

Check weaknesses.

---

> ### Author Response · Authors · 2025-11-24
>
> We thank the reviewer for their comments. We address each of the reviewer’s concerns below:
>
> > I find the eval for generative LLMs limited and biased: The head-to-head with GPT-4o / o3-mini uses only 100 test cells per dataset and constrains the label space to a provided list. That yields to sample-inefficient and risks optimistic estimates (especially for few, separable labels).
>
> Our results involving generative LLMs and a constrained label space are not optimistic relative to scGPT, since scGPT uses k-nearest neighbours for cell type classification which naturally has a constrained label space in that it will also only output labels present in the train set. This is discussed in Appendix A.3.1. Furthermore, while we use 100 cells per dataset, we note that we report results on 3 different datasets, and conduct 3 runs on each dataset, reporting mean and standard deviation. Therefore, our results are robust in showing statistical significance. Evaluating on the entire dataset with this setup where we use multiple datasets and multiple runs on each dataset would be prohibitively expensive.
>
> > If I follow correctly the methodology, IG and LIME are applied to an MLP on frozen Ember-V1 embeddings, not to the end-to-end LLM or the generative-LLM setup. So attributions correspond the MLP+Ember composite, not to scMPT nor the reranking LLMs.
>
> This is correct. However, note the overarching goals of our study, which are: 1) To investigate what biological insights contribute to the performance of LLMs when applied to single-cell data, and 2) To investigate how these models can complement single-cell foundation models to improve upon their performance. The interpretability analysis in this study focusing on Ember-V1+MLP provides stronger evidence for goal 1), which is what it is intended to contribute towards, by focusing specifically on the LLM.
>
> > My main concern is that the ablations might hint some shortcut learning: Hashing names wipes performance, which might mean heavy reliance on lexical identity rather than biological semantics. These patterns look like lexical+rank shortcuts, not robust biology. The authors should evaluate under batch shift and low-depth cells.
>
> We would first like to note that we show the models do leverage knowledge of biology through 1) Our interpretability tests.
> 2) Through showing that gene name embeddings for marker genes of the same cell type are on average more similar than embeddings of marker gene embeddings for different cell types.
> 3) Through our ablation tests, which show a moderate drop in performance when hashing gene names.
> With that said, we agree with the reviewer that the reliance on lexical similarity and simple gene rank is a potential limitation of LLMs, and it may affect generalizability. Generalizability could be interesting to investigate in future work - but additional experiments could result in scope creep for our already extensive study. We thank the reviewer for pointing this out, however, and will discuss this limitation of LLMs in our expanded discussion section. We further believe that our work showing this limitation of LLMs increases its potential for impact, especially given recent interest in applying these models to single-cell analysis.
>
> > Regarding the claims about the knowledge of marker genes, we observe only some correlational evidence. In order to assure such claim, a stronger causal probe would be necessary, for example through counterfactual token tests.
>
> The evidence from our interpretability methods is correlational relative to the underlying data, but not relative to the model. For instance, LIME involves training an interpretable model on top of a dataset consisting of perturbed samples and the corresponding predictions from the model of interest. In this way, investigating the effect of varying data on the model’s predictions, or from a causal perspective, investigating the effect of intervention (see: https://christophm.github.io/interpretable-ml-book/lime.html for more details). Note that our goal with this experiment is not to show that “expression of GCG causes a cell to be an alpha cell”, but to investigate how the presence of certain genes influences the model’s predictions. Our interpretability methods accomplish this, and our investigation of the concordance between interpretability methods strengthen our findings.
>
> We hope our rebuttal addresses the reviewer’s concerns and clarifies our methodology. If the reviewer is satisfied with our response, we would appreciate it if they would consider revising their assessment of our work. We are also happy to engage in further discussion.

---

> > ### Author Response · Authors · 2025-11-28
> >
> > We would like to thank the reviewer F1vo once again for their comments, and their time spent reviewing our work. Since the end of the discussion period is quickly approaching, we would like to know if there are any remaining questions, suggestions, or concerns so that we can have a chance to address them. We note that in our rebuttal, we have provided clarifications that address each of the reviewer's concerns. If our rebuttal is satisfactory in addressing these concerns, we would appreciate if the reviewer would consider revising their assessment of our paper. We thank the reviewer once again for their time.

---

### Official Review · Reviewer_7brm · 2025-10-31

**Soundness:** 3
**Presentation:** 3
**Contribution:** 2
**Rating:** 4
**Confidence:** 4

**Summary:**

Summary
This paper (Bridging Gene Expression and Text: LLMs Can Complement Single-Cell Foundation Models) investigates how large language models (LLMs) can complement single-cell foundation models (scFMs) such as scGPT for cell type classification and related single-cell analysis tasks. The authors systematically evaluate several LLMs (e.g., Ember-V1 for encoder, GPT-4o for non-reasoning LLM, o3-mini for reasoning model) on cell sentence representations (cell represented as sorted genes, commonly use in most LLM based cell foundation models), perform interpretability and ablation studies, and propose scMPT, a simple fusion model that concatenates scGPT and LLM embeddings followed by an MLP classifier. Results show modest improvements over individual models and qualitative evidence that LLMs capture marker-gene knowledge and simple gene-expression patterns.

Overall Assessment
This work provides a careful and systematic evaluation of how LLMs interact with scFMs and contributes valuable insights for future research. However, the methodological innovation is limited, the improvements are small and insufficiently substantiated, and the experiments focus on relatively easy tasks.

**Strengths:**

1. The paper is well written and clearly structured, with strong motivation for exploring complementarity between text-based LLMs and single-cell foundation models.

2. Provides comprehensive, systematic experiments including ablations, interpretability analyses (integrated gradients and LIME), and comparisons across datasets to explore what the LLMs use to perform the cell-oriented classification tasks. This is a very good angle and offering a fresh perspective for the community.

3. The concept of using LLMs as complementary rather than alternative to scFMs is timely

4. The work contributes useful empirical baselines for future multimodal single-cell modeling studies.

**Weaknesses:**

1. Limited architectural novelty.
The proposed fusion model (scMPT) concatenates scGPT and LLM embeddings followed by an MLP. This is a straightforward late-fusion ensemble rather than a novel modeling approach. The contribution is primarily empirical, focusing on evaluating existing models on existing datasets.

2. Use only scGPT to represent scFMs.
There are many other scFMs (e.g. scBERT, scPRINT, scFoundation, geneformer, just to name a few) which represent the cell in different ways, and may interact with LLM differently. Using only 1 seems relatively weak.

3. Small dataset
The generative re-ranking experiment uses only 100 samples per dataset, which is too few to claim meaningful cost savings or statistical significance.

4. Over-reliance on simple tasks.
The primary evaluation task is cell type classification, is known to be relatively easy (especially for well known cell types), as marker genes alone yield high accuracy. The study does not address more challenging cases such as fine-grained subtypes or rare cell populations, where complementary multimodal reasoning would be more revealing.

5. Scope dilution.
Although interesting, the inclusion of LLM-based reasoning experiments (e.g., scGPT + o3-mini / DeepSeek-R1) feels underdeveloped and somewhat disconnected from the main contribution. The small scale and lack of deeper analysis make it difficult to interpret their significance (e.g. o3-mini alone outperform other models in the Pancreas dataset).

6. Unrealized potential of the fusion idea.
The fusion concept is promising, but a more principled hybrid approach, such as alignment / contrastive pretraining between scGPT and LLM embeddings, could yield stronger gains. The current setup does not fully exploit cross-modal synergies.

**Questions:**

1. Can the authors show per-class performance, especially for rare or ambiguous cell types?

2. Have the authors considered alternative fusion strategies (e.g., gating, cross-attention, contrastive alignment)?

**Details Of Ethics Concerns:**

No concern.

---

> ### Author Response · Authors · 2025-11-24
> **(1/2)**
>
> We thank the reviewer for their comments, and for recognizing that our work contributes valuable insights for future research in this space. We address the reviewer’s concerns below:
>
> >Use only scGPT to represent scFMs. There are many other scFMs (e.g. scBERT, scPRINT, scFoundation, geneformer, just to name a few) which represent the cell in different ways, and may interact with LLM differently. Using only 1 seems relatively weak.
>
> We do not only use scGPT to represent scFMs, we use Geneformer as well. This is discussed in Section 4.2, with results in Table 21 and 22 of Appendix A.2.
>
> >Small dataset The generative re-ranking experiment uses only 100 samples per dataset, which is too few to claim meaningful cost savings or statistical significance.
>
> While we use 100 cells per dataset, we note that we report results on 3 different datasets, and conduct 3 runs on each dataset, reporting mean and standard deviation. Therefore, our results do show statistical significance and are not underdeveloped. Evaluating on the entire dataset with this setup where we use multiple datasets and multiple runs on each dataset would be prohibitively expensive.
>
> > Over-reliance on simple tasks. The primary evaluation task is cell type classification, is known to be relatively easy (especially for well known cell types), as marker genes alone yield high accuracy. The study does not address more challenging cases such as fine-grained subtypes or rare cell populations, where complementary multimodal reasoning would be more revealing.
>
> The cell type classification datasets we used were a meaningful challenge for single-cell foundation models and LLMs given that on several datasets (ie. Myeloid, Bones, MS) performance was quite poor (see Appendix A.1). We note that while Table 1 in the main body of our paper, which corresponds to the Pancreas dataset, contains cell types that are quite general and well known (ie. Alpha cells), other datasets contain more fine-grained subtypes. For instance, the list of cell types in the MS dataset is as follows: ['cortical layer 5-6 excitatory neuron', 'PVALB-expressing interneuron', 'cortical layer 2-3 excitatory neuron B', 'oligodendrocyte C', 'VIP-expressing interneuron', 'pyramidal neuron', 'cortical layer 4 excitatory neuron', 'SV2C-expressing interneuron', 'cortical layer 2-3 excitatory neuron A', 'SST-expressing interneuron', 'mixed glial cell', 'mixed excitatory neuron', 'oligodendrocyte precursor cell', 'astrocyte', 'microglial cell', 'endothelial cell', 'oligodendrocyte A', 'phagocyte']. The cell types in the Bones dataset are: ['RepC' (reparative chondrocytes), 'preHTC’ (prehypertrophic chondrocytes), 'RegC' (regulatory chondrocytes), 'HomC' (homeostatic chondrocytes), 'HTC’ (hypertrophic chondrocytes), 'preFC' (prefibrochondrocytes), 'FC' (fibrochondrocytes)]. Lastly, we note that we do not only evaluate cell-type classification, and also evaluate disease phenotype prediction performance.
>
> > Limited architectural novelty. The proposed fusion model (scMPT) concatenates scGPT and LLM embeddings followed by an MLP. This is a straightforward late-fusion ensemble rather than a novel modeling approach. The contribution is primarily empirical, focusing on evaluating existing models on existing datasets.
>
> The architectures used in our work are not intended to be novel. The simplicity of our approach (ie. scMPT) is intentional and strengthens our results, in that its strong performance highlights that representations from LLMs and single-cell foundation models are truly synergistic (in contrast, a more complex architecture may exhibit better performance simply because of its higher capacity). However, while scMPT is simple, combining single-cell representations from LLMs and single-cell foundation models is a novel approach and our work is not simply “evaluating existing models on existing datasets”.
>
> > Have the authors considered alternative fusion strategies (e.g., gating, cross-attention, contrastive alignment)?
>
> We considered cross-attention, but early results did not show substantial performance improvements. We agree with the reviewer that more sophisticated approaches such as contrastive alignment would be interesting to explore. However, such sophisticated approaches go beyond the scope of this work, where our objective was to explore what knowledge of biology is captured in LLM generated representations of single-cell data, and investigate whether these representations are synergistic with representations generated by single-cell foundation models. Our work, which is already quite extensive, provides a valuable foundation for future work to explore these more sophisticated approaches.

---

> > ### Author Response · Authors · 2025-11-24
> > **(2/2)**
> >
> > > Can the authors show per-class performance, especially for rare or ambiguous cell types?
> >
> > We investigated per-class accuracy, but found that the performance of an encoder on identifying a given cell type is dataset dependent. For instance, as shown below, Ember-V1 is more accurate than scGPT at identifying beta cells in the Pancreas dataset, whereas scGPT is much more accurate than Ember-V1 at identifying beta cells in the Tabula Sapiens dataset. The use of k-nearest neighbours for zero-shot cell type classification could contribute toward this. In k-nearest neighbours, a given cell type having poor embeddings can influence the prediction accuracy not only of that cell type, but of other cell types, making it difficult to extract meaningful conclusions from inspecting per-class accuracy. However, we note that k-nearest neighbours is widely used in related work, so this limitation is not specific to our study.
> >
> >
> > Per Class Accuracy- Pancreas Dataset:
> >
> > | Class         | scGPT| Ember-V1 |
> > |--------------|--------|--------------|
> > | MHC class II | 0.00   | 0.00         |
> > | PP           | 0.32   | 0.99         |
> > | PSC          | 0.78   | 0.90         |
> > | acinar       | 0.62   | 0.78         |
> > | alpha        | 0.89   | 0.99         |
> > | beta         | 0.94   | 0.99         |
> > | delta        | 0.79   | 0.96         |
> > | ductal       | 1.00   | 0.97         |
> > | endothelial  | 1.00   | 1.00         |
> > | epsilon      | 0.14   | 0.71         |
> > | mast         | 0.00   | 0.71         |
> >
> >
> > Per Class Accuracy - Tabula Sapiens Dataset (Pancreatic Cells):
> >
> > | Class                       | scGPT | Ember-V1|
> > |----------------------------|--------|--------------|
> > | pancreatic acinar cell     | 0.98   | 0.93         |
> > | pancreatic alpha cell      | 0.00   | 0.00         |
> > | pancreatic beta cell       | 0.71   | 0.29         |
> > | pancreatic delta cell      | 0.00   | 0.00         |
> > | pancreatic ductal cell     | 0.73   | 0.65         |
> > | pancreatic pp cell         | 0.00   | 0.00         |
> > | pancreatic stellate cell   | 0.50   | 0.50         |
> >
> >
> > If the reviewer is satisfied with our response, we would appreciate it if they would consider revising their assessment of our work. We are also happy to engage in further discussion.

---

> > > ### Author Response · Authors · 2025-11-28
> > >
> > > We would like to thank the reviewer 7brm once again for their comments, and their time spent reviewing our work. Since the end of the discussion period is quickly approaching, we would like to know if there are any remaining questions or concerns so that we can have a chance to address them. We would like to note that in our rebuttal, we have addressed each of the reviewer's concerns, including clarifying that we do incorporate another single-cell foundation model (Geneformer) into our experiments, and that our experiments with generative models involve repeated trials, and therefore do show statistical significance. If our rebuttal is satisfactory in addressing the reviewer's concerns, we would appreciate if the reviewer would consider revising their assessment of our paper. We thank the reviewer once again for their time.

---

### Official Review · Reviewer_MJWZ · 2025-11-06

**Soundness:** 2
**Presentation:** 3
**Contribution:** 2
**Rating:** 4
**Confidence:** 3

**Summary:**

This paper studies what biological insights contribute toward the performance of LMs when applied to single-cell data, and how these models can complement single-cell foundation models to improve upon their performance.

For the former, they found that LMs capture biological insight, and specifically knowledge of marker genes, as well as simple but effective gene expression pattern (top marker gene patterns).

For the later, they introduced scMPT, which leverages both the representations generated by scGPT and an Ember-V1 text encoder, enabling better overall performance for cell-type classification and disease phenotype prediction.

**Strengths:**

- The paper tackles a relevant and timely question in computational biology: how language models interpret gene-level signals in single-cell data, and how this can be leveraged to complement single cell specialized foundation models.

- Interesting findings and analysis: The interpretability analysis connecting marker genes with model attributions is interesting and may be useful to biological researchers. And the discussion on gene-name hashing raises a thoughtful point—why language models might already capture much of the relational information between genes through their co-occurrence in “cell sentences,” rather than through explicit gene-name semantics.

- The paper is clearly written.

**Weaknesses:**

- Recommend for journal submission instead of ML conference: The method itself does not introduce substantial theoretical or engineering innovations. It resembles an empirical benchmark or ablation study for a biological problem rather than a new modeling framework.

- Limited benchmarking: The experiments include a narrow range of datasets and baselines. Since scGPT’s release, many other cell foundation models have emerged, and traditional non-foundation approaches remain competitive. Without broader comparisons, it is difficult to establish whether scMPT achieves state-of-the-art performance.

- Weak ML insight: While the biological analysis is sound, combining two frozen encoders and concatenate the two features for classification sounds like years-old architecture. Or in other words, the paper does not provide much new ML insights, such as novel objectives, architectures, or theoretical findings.

**Questions:**

1. how is the model in Figure 1 is trained? Directly using cell type classification using cross entropy loss to train from scratch? If yes, then is it natural to observe the interpretability methods give high score to marker genes as explanation? Because that’s the determinant part for cell types, and cell types are used for supervision.

2. Could you clarify why gene-name hashing improves performance, based on your understanding? Do you believe semantic information from gene names contributes meaningfully beyond their co-occurrence statistics in cell contexts?

3. Will you expand benchmarking to include additional more advanced cell foundation models and traditional baselines (e.g., scANVI)?

---

> ### Author Response · Authors · 2025-11-24
>
> We thank the reviewer for their comments, and for recognizing that the problem our work focuses on is important, and that our findings are interesting and have potential to meaningfully benefit the computational biology community. Below we address the questions and concerns provided:
>
> > Since scGPT’s release, many other cell foundation models have emerged, and traditional non-foundation approaches remain competitive. Without broader comparisons, it is difficult to establish whether scMPT achieves state-of-the-art performance.
>
> We do not compare scMPT to non-foundation model baselines because the goal of scMPT is not to develop a state of the art architecture - it is to show that LLM and single-cell foundation model representations are synergistic. We would also like to note that we do incorporate another single-cell foundation model (Geneformer- as discussed in section 4.2) in our experiments to show our findings are generalizable to single-cell foundation models beyond scGPT.
>
> > While the biological analysis is sound, combining two frozen encoders and concatenate the two features for classification sounds like years-old architecture
>
> The architectures used in our work are not intended to be novel. The simplicity of our approach (ie. scMPT) is intentional, in that its strong performance highlights that representations from LLMs and single-cell foundation models are truly synergistic (in contrast, a more complex architecture may exhibit better performance simply because of its higher capacity). We would also like to note that the reviewer guidelines stress the value of empirical knowledge that is useful to practitioners in the acceptance/rejection criterion related to novelty (see “Reviewing a submission: step-by-step” section 3  https://iclr.cc/Conferences/2026/ReviewerGuide).
>
> > how is the model in Figure 1 is trained? Directly using cell type classification using cross entropy loss to train from scratch? If yes, then is it natural to observe the interpretability methods give high score to marker genes as explanation? Because that’s the determinant part for cell types, and cell types are used for supervision.
>
> scMPT was trained on cell type classification. Note, however, that interpretability methods were actually used on the Ember-V1+MLP setup, but this model was trained on cell type classification as well. In both cases though, the models are not trained from scratch - the encoders are left frozen, which partially addresses the reviewer’s concern about knowledge of marker genes potentially being derived from training on cell type classification. However, we acknowledge that this does not fully address the reviewer’s concern as there is still a small trainable MLP component. We mitigate this limitation, however, by providing other evidence that the model leverages parametric knowledge of biology - specifically, we show that gene name embeddings for marker genes of the same cell type are on average more similar than embeddings of marker gene embeddings for different cell types (see our discussion section). The moderate drop in performance when ablating gene names also provides evidence of the models leveraging knowledge of biology obtained from training on general text data, as otherwise we would expect to see a smaller drop.
>
> > Could you clarify why gene-name hashing improves performance, based on your understanding?
>
> Gene name hashing did not improve performance. Performance in the “Gene Name Ablation” column of Table 2, and the tables in Appendix A.4 was overall worse than performance in the Baseline column.
>
> We hope our rebuttal addresses the reviewer's concerns. If the reviewer is satisfied with our response, we would appreciate if they would consider revising their assessment of our work. We are also happy to engage in further discussion.

---

> > ### Author Response · Authors · 2025-11-28
> >
> > We would like to thank the reviewer MJWZ once again for their comments, and their time spent reviewing our work. Since the end of the discussion period is quickly approaching, we would like to know if there are any remaining questions or concerns so that we can have a chance to address them. We would like to note that we provided answers to the reviewer's questions, and clarifications for the reviewer's concerns, including clarifying that we do incorporate another single-cell foundation model (Geneformer) into our experiments. If our rebuttal is satisfactory in addressing the reviewer's concerns, we would appreciate if the reviewer would consider revising their assessment of our paper. We thank the reviewer once again for their time.

---

### Author Response · Authors · 2025-12-03
**Summary comment for AC**

We sincerely thank the reviewers for their time spent reviewing our work, their comments, and their feedback. We are happy to see that reviewers recognize our work’s potential to be useful to researchers in our field (MJWZ, 7brm), that they find our interpretability analysis is interesting and provides solid evidence to support our claims (MJWZ, 7brm, F1vo, 615z), and that our overarching question of how language models can be leveraged to complement single-cell foundation models is timely and well motivated (MJWZ, 7brm).

We have responded to all of the reviewers’ questions and concerns, and are disappointed the discussion period was prematurely closed to the reviewers before they could respond. Our rebuttal addresses several major misunderstandings related to our methodology, and the objectives of our study. We provide three illustrative examples below:

1. Reviewers MJWZ and 7brm criticize our work for only including one single-cell foundation model (scGPT), however, we also experiment with Geneformer as an alternate single-cell foundation model (which we discuss in Section 4.2).
2. Reviewer 615z claims that a “critical flaw” in our methodology is that we use an easier train/test split than the scGPT paper on the human pancreas dataset, highlighting that this was their “most important concern”. However, we use the same split as scGPT, which we discuss in our rebuttal.
3. Several reviewers (MJWZ, 7brm) misunderstood the intended contribution of our scMPT proof-of-concept model, criticizing it for its architectural simplicity. However, scMPT is not intended to be an architecturally novel state-of-the-art model, and this simplicity is intentional, in that the model’s strong performance highlights that representations from LLMs and single-cell foundation models are truly synergistic (in contrast, a more complex architecture may exhibit better performance simply because of its higher capacity). Furthermore, combining single-cell representations from LLMs and single-cell foundation models is a novel approach in an underexplored area, which ultimately provided valuable insights, and this experiment is not simply “evaluating existing models on existing datasets” as claimed by reviewer 7brm.

We respond to several other misunderstandings in our rebuttal, and we thank the AC for their time in reviewing and considering these responses, especially given the recent changes to the rebuttal/discussion period. We note that we have also edited the discussion section of our paper to address reviewer feedback (from 615z, F1vo).

---

### Meta-Review · Area_Chair_viPg · 2026-01-05

**Summary:**

Reviewers had overall weakly to strongly negative initial reviews. The primary leading to my suggestion to reject the paper in its current form are based on the remarks by reviewers and my own read of the paper.

1. Weak "alignment" with ML conference, focusing more on empirical benchmarking of a biological problem (reviewer `MJWZ`).
2. In a similar vein to the the point above: Limited architectural novelty / ML insights (reviewer `7brm`).
3. Missing analysis of robustness under batch effects (reviewer `F1vo`).
4. Potential methodological/evaluation issues (reviewer `615z`).

Looking at the initial reviews and discussion, I find myself agreeing with some of the points made by reviewers `MJWZ` and `7brm`, viz., that in its current state, the submission should rather be evaluated based on its biological merits rather than its ML merits. This necessitates a different set of reviewers and a better focus on the biological insights, though; the current set of reviewers all have a strong ML background with _some_ experience in (computational) biology, but their reactions indicate that there is a misalignment between the intended audiences. As such, and based on the other concerns raised above, I cannot endorse this submission for publication at the moment.

**Reviewer Concerns:**

For reviewer `MJWZ`:

- Concerns about the training setup have been successfully addressed by the rebuttal.
- Concerns about the novelty have _not_ been addressed by the rebuttal. While the authors are right that empirical results that are relevant for practitioners should have a space at such conferences, the reviewer guide considers practitioners to be _ML_ practitioners; as it stands, the intended audience of this paper appears to be a biological one. Here, the authors correctly focus on _simplicity_ over elaborate architectures (which is commendable!), but I do not find that the rebuttal can address the architectural/novelty concerns sufficiently (see my comment on the "misalignment of audiences" above).

For reviewer `7brm`:

- Concerns about limited architectural novelty have not been addressed by the rebuttal (see comment above).
- Additional queries about per-class performance have been well addressed by the rebuttal.
- Concerns about the statistical significance assessed over small datasets have been _partially_ addressed in the rebuttal. While I agree with the authors that the repetitions they used lead to statistically significant results, a comment made on computational feasibility raises some additional concerns on the practical applicability of the method, thus indicating the need for an additional assessment by reviewers highly familiar with the problem domain.

For reviewer `F1vo`:

- Concerns about biases in evaluation could be addressed by the authors.
- The concerns raised about batch effects have not been addressed satisfactorily in the rebuttal. The request for analyzing such a crucial issue is not "scope creep," as claimed by the authors, but a _central_ form of ablation required for these type of analyses.

For reviewer `615z`:

- Concerns about missing baselines could be partially addressed by the authors.
- Concerns about the dataset selection procedure could be addressed by the authors.
- Concerns about some of the claims and their phrasing could be partially addressed by the authors.
- The overall concerns about (potential) methodological issues appear to be unaddressed, though.

**Reviewer Scores:**

- `MJWZ`: Initial score 4, estimated final score 4 (given that their concerns on novelty cannot be easily addressed)
- `7brm`: Initial score 4, estimated final score 4 (given that their concerns on architectural / ML novelty cannot be easily addressed)
- `F1vo`: Initial score 4, estimated final score 6 (given that many concerns have been successfully addressed)
- `615z`: Initial score 2, estimated final score 4 (given that the reviewer appears concerned about methodological issues)

It is unlikely that any of the reviewers would have decided to "champion" the paper; I find their feedback to be useful and actionable, insofar as it points towards additional venues for publication. I realize that this is not the desired outcome for the authors, but I nevertheless hope that they can make use of the comments.

---

### Decision · Program_Chairs · 2026-01-26

Reject